# CRITIC-COT: BOOSTING THE REASONING ABILITIES OF LARGE LANGUAGE MODEL VIA CHAIN-OF-THOUGHT CRITIC

## ABSTRACT

Self-critic has become a crucial mechanism for enhancing the reasoning performance of LLMs. However, current approaches mainly involve basic prompts for intuitive instance-level feedback, which resembles System-1 processes and limits the reasoning capabilities. Moreover, there is a lack of in-depth investigations into the relationship between LLM's ability to criticize and its task-solving performance. To address these issues, we propose Critic-CoT, a novel framework that pushes LLMs toward System-2-like critic capability. Through a step-wise CoT reasoning paradigm and the automatic construction of weak-supervision data without human annotation, Critic-CoT enables LLMs to engage in slow, analytic self-critique and refinement, thereby improving their reasoning abilities. Experiments on GSM8K and MATH demonstrate that our enhanced model significantly boosts task-solving performance by filtering out invalid solutions or iterative refinement. Furthermore, we investigate the intrinsic correlation between critique and task-solving abilities within LLMs, discovering that these abilities can mutually reinforce each other rather than conflict.

## 1 INTRODUCTION

Enhancing the reasoning abilities of large language models is essential for creating more intelligent and reliable AI systems, which has drawn extensive attention from researchers (Chollet, 2019; Bubeck et al., 2023; Morris et al., 2024). From a cognitive perspective, the procedure of human reasoning involves constant reflection and revision (Hegel et al., 1991; Kierkegaard, 1989; Popper, 1934), which has inspired increasing focus on integrating self-critic mechanisms in the reasoning process of large-scale models (Kim et al., 2023; Shinn et al., 2023; Madaan et al., 2023). This involves iteratively allowing the model to generate feedback on its own responses and then refining its reasoning based on the feedback. Compared with traditional critic methods that depend on feedback from external sources (Saunders et al., 2022; McAleese et al., 2024), self-critic relies solely on the model's internal capabilities, thus reducing the high cost of additional human annotation, and serving as a promising potential solution to scalable oversight (Leike et al., 2018; Burns et al., 2023; Cao et al., 2024).

However, current studies primarily focus on utilizing LLMs' critique abilities to enhance their performance. Yet, relatively little attention has been given to the investigation and development of the critique ability itself. Firstly, existing critique methods are often overly simplistic, typically relying on a basic prompt to directly point out the error, without stepwise Chain-of-Thought examination or training procedure, which leads to relatively poor self-critic accuracy (Luo et al., 2023; West et al., 2024). Specifically, proposing a valid critique is a complicated task that requires a thorough understanding of statements and precise negativity. However, current LLMs are normally not explicitly trained for critic capability. Therefore, these simple approaches usually tend to "criticize" like System-1, which is more intuitive and likely to make mistakes, rather than more rigorous and deliberate System-2 (Kahneman, 2011; Yu et al., 2024), while shifting LLMs from System-1 toward System-2 emerges as a promising approach for improving the reasoning capability (OpenAI, 2024). This limitation diminishes the effectiveness of self-critic and, further, self-correct (Huang et al., 2024). Secondly, the capabilities of task-solving and self-critic are both dependent on the model's inherent knowledge, while there is currently a lack of in-depth exploration regarding the

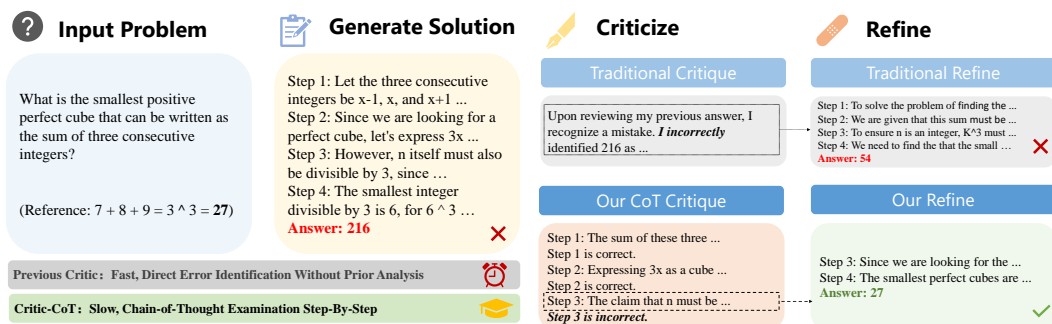

Figure 1: Illustration of Critic-CoT: Previous instance-level critic methods attempt to identify errors directly without any prior analysis, and restart from the beginning during refinement. In contrast, our proposed Critic-CoT framework performs a step-wise examination using the Chain-of-Thought approach. When refining, rather than starting from scratch, our method makes the correction from the specific problematic step with the help of the corresponding critique.

correlation between these two capabilities within LLMs. In that case, it's challenging to balance the task-solving and the self-critic capabilities of the model within the self-critic framework, which poses a significant obstacle to the subsequent development in this direction.

To this end, this paper is devoted to diving into the following critical research questions:

- How can we enhance a model's critique ability, pushing it toward System 2 reasoning?

- What is the relationship between a model's critique ability and its task-solving capability?

To answer the above questions, as shown in Figure 1, we propose Critic-CoT, a novel framework designed to enhance LLMs' reasoning abilities. Through step-wise Chain-of-Thought critique format and weak supervision, our method is able to strengthen System-2-like critic ability, without the intensive cost of human annotation. Specifically, during training, we let LLMs criticize and refine their solutions in a complete CoT way, and collect successful pairs that convert wrong solutions into correct ones, or affirm the validity of original right solutions. After supervised fine-tuning on the obtained step-wise critic-refine data, we enable the target LLM to analyze and criticize each step of its generated reasoning procedure, so that it can filter out wrong attempts and preserve the correct ones with greater precision. During inference, to leverage the model's abilities of CoT-critique and refinement, we employ two strategies: (1) majority vote filtering involves using the critic model to evaluate multiple generated solutions and filter out those incorrect; and (2) iterative refinement, on the other hand, involves repeatedly critiquing and refining a solution until no further error is detected.

Through a series of experiments on the dataset of GSM8K (Cobbe et al., 2021a) and MATH (Hendrycks et al., 2021), we found that our trained critic model can fairly distinguish incorrect solutions from correct ones, and improve the reasoning accuracy via iterative refinement or critic filtering. These results demonstrate the helpfulness and effectiveness of our proposed method. Additionally, we observed that our critic model already exhibits noticeable performance improvements in task-solving, even in the absence of additional critique steps during the decoding phase. Such findings reveal that strengthening the ability to critique and refinement would not compromise the task-solving performance, but improve it. This also suggests the presence of an intrinsic mechanism by which critique ability and task-solving capability mutually reinforce one another.

We summarize our main contributions as follows:

- We propose Critic-CoT, which pushes the critic paradigm of LLMs from System-1-like incentive "thinking" toward System-2-like deliberate "reasoning".

- Through experiments, we find that Critic-CoT can effectively teach the model to criticize and refine its own output step by step, thus noticeably improving the reasoning performance.

- Moreover, we find that for LLMs, the ability of critique and refinement could mutually reinforce, which may shed light on designing more advanced self-critic framework designs in future work.

## 2 RELATED WORKS

### 2.1 DISCRIMINATIVE VERIFIER FOR MATHEMATICS

To further improve the reasoning ability of Large language models, one applicable approach is through the use of reward models, which can either be used in reinforcement learning during training (Ouyang et al., 2022) or rejection sampling at test time (Cobbe et al., 2021b). While outcome-supervised reward models (ORMs) allow for the automatic collection of training data based on the signal of the gold answer, process-supervised reward models (PRMs) would be more advantageous for more precise feedback, better interpretability and stronger alignment (Lightman et al., 2024).

To reduce the considerable human labeling cost and difficulty for dense annotation, a series of works based on automatic approaches have been proposed (Wang et al., 2023a; Chen et al., 2024b; Luo et al., 2024; Snell et al., 2024), all under the heuristic that for an incorrect solution, the first error step is where the continuation of previous step would lead to a correct answer. This may bring noise into training data due to false positives and negatives (Luo et al., 2024). Moreover, annotation based on the implicit solution continuation alone does not leverage LLM's emerging ability of critic, which is in a more explicit and analytic way and brings better explainability (Saunders et al., 2022; Yuan et al., 2024; Luo et al., 2023; McAleese et al., 2024). Additionally, binary 0/1 discrimination alone, whether outcome-based or process-based, remains more similar to System-1 reasoning rather than the desirable System-2, thus may not fully leverage the computation power support by empirically successful Chain-of-Thought prompting (Feng et al., 2023; Li et al., 2024).

### 2.2 CRITIC MODEL

Learning from natural language feedback could be beneficial (Chen et al., 2024a). With the development of LLM, whether it can discriminate and criticize its own output in a text-generation manner becomes an interesting topic (Luo et al., 2023; Zeng et al., 2023), with doubts at least on off-the-shelf LLMs that are not specially trained for such task (Huang et al., 2024; West et al., 2024; Liu et al., 2024). Later, several works are proposed to improve self-reflection via a carefully designed prompting pipeline (Zhang et al., 2024b; Yan et al., 2024; Wu et al., 2024). Current applications, such as response evaluation, heavily rely on the reference (Zheng et al., 2023). Therefore, given the limited critic ability of current LLMs, how to train a robust and applicable critic model is worth investigating. Concurrently, Zhang et al. (2024a) trained a generative reward model on the outcome level rather than the process level but did not incorporate refinement into the schema.

From the perspective of recursive reward modeling (Leike et al., 2018; Saunders et al., 2022) and scalable oversight (Burns et al., 2023), McAleese et al. (2024) recently trained "CriticGPT" to assist human labelers, which aims to improve the ability of human rather than the base model, i.e. improve the overall recall of error detection, rather than precision. While in this paper, we try to explore whether improving the reasoning ability of LLM without costly human annotation is applicable.

## 3 METHOD

To equip LLMs with the ability to criticize and refine themselves step-by-step, we propose Critic CoT. As shown in Figure 2, it consists of two modules, including weak-supervision-based auto-train and self-check at inference-time. First, we introduce the weak-supervision principles in Section 3.1, followed by the training process in Section 3.1, and finally, the inference strategies in 3.3.

### 3.1 CHAIN-OF-THOUGHT CRITIQUE

In this work, we utilize a step-wise chain-of-thought critique, which makes the critique-refine process both controllable and formalizable, thereby facilitating the collection of weak supervision data. Formally, given the question $Q$ and the corresponding gold answer $Ans$, we have the $n$-step attempt

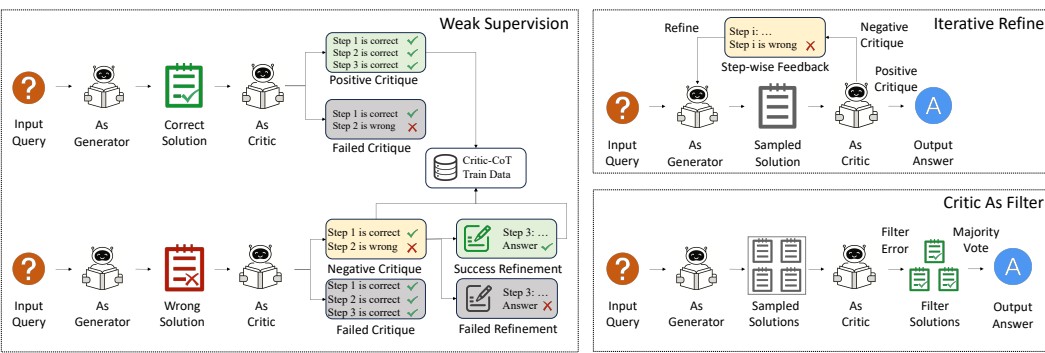

(a) Auto-Train: Training data construction of Critic-CoT

(b) Self-Check: Inference process of Iterative Refine and Critic As Filter

Figure 2: The Process of Critic-CoT during training (a) and inference (b). For training, we collect the critic-refine data on the generator's samples via weak supervision (Section 3.1). Through fine-tuning, we enable the target model to criticize and refine its own reasoning process. Then, during inference, we can leverage the capabilities via Iterative Refine or Critic As Filter (Section 3.3).

$Att = [s_1, ..., s_n]$ with predicted answer $Pred$ sampled by generator $G$. The corresponding critique $Cri$ then can be represented as $L = [l_1, ..., l_n]$, where the step label $l_i = +1$ indicates that step $i$ is predicted to be correct, and $l_i = -1$ to be incorrect. Then the refinement $Att' = [s'_i, ..., s'_{n'}]$ is start from the first incorrect step $i$ with new answer $Pred'$. To automatically annotate the process label for the attempts, we assume that (1) If the final answer is wrong, then there is one earliest mistake, and by refining from this mistake, we could reach a correct answer; (2) If the final answer is correct, then all the intermediate steps are correct. Thus, we enumerate the following cases:

- $Pred \neq Ans, -1 \notin L$: The attempt is wrong, yet the critique did not discover any error step. Thus the critique itself is problematic, and we need to sample another critique.

- $Pred \neq Ans, -1 \in L, Pred' \neq Ans$: The attempt is wrong, and the critique found an error, but still, the refinement is not correct. There could be two cases for this situation: (1) the refinement is unsuccessful; (2) the critique did not detect an earlier mistake. We simply sample another critique and corresponding refinement for this situation.

- $Pred \neq Ans, -1 \in L, Pred' = Ans$: Not only did the critique point out the error, but also the refinement reached the correct answer. We then believe the critique is valid, and collect the critique data instance $C = (Q, Att, Cri)$ and the refinement data $R = (Q, Att, Cri_{-1}, Att')$, where $Cri_{-1}$ is the critique of last step, since explaining why previous steps are correct may not be helpful for refinement.

- $Pred = Ans, -1 \notin L$: The attempt is correct, and the critique believes it is correct. So we can collect the positive critique data instance $C = (Q, Att, Cri)$.

- $Pred = Ans, -1 \in L$: The attempt reached the correct answer, yet the critique found an error. Then, the critique could be wrong, and we need to sample another critique.

## 3.2 AUTO TRAIN: TWO-STAGE TRAINING

To enable the model to acquire self-critiquing and refining capabilities, we first need to provide it with basic critiquing abilities, followed by self-critique for further enhancement. The overall training procedure is divided into two stages.

**Stage 1** In the first step, we collect high-quality critique data to provide the model's basic critiquing ability. Specifically, we first sample both positive and negative solutions from a representative instruction-following model $\mathcal{M}_G$ on the dataset $D$. Then, we utilize SOTA LLMs like GPT4-Turbo to serve as critic model $M_C$. For each generated attempt $Att$, the critic model will retry at most $k$ times to produce a valid critique until it reaches one of the weak supervision constraints. This will form the critic-refine dataset $D_1 = \{(Q, Att, Cri)\} \bigcup \{(Q, Att, Cri_{-1}, Att')\}$ for fine-tuning the initial model $\mathcal{M}_0$ into the critic model $\mathcal{M}_1$. Note that in this process, we actually distill

`Pass1@N` of the teacher model $\mathcal{M}_C$ into `Top1@N` of the student model. So, the theoretical upper bound of the student model is not necessarily limited by the teacher model's performance.

**Stage 2**  In the second step, we leverage the model's self-critique to enhance its critiquing and refining capabilities further. Namely, we let the learned critic model $\mathcal{M}_1$ criticize and refine its own output. We first sample $M$ correct-answer solutions and $M$ incorrect-answer solutions for each question $Q$ in the original dataset $D$. Then, for each attempt $Att$, we employ $\mathcal{M}_1$ to repeatedly criticize and refine at most $k$ times. In case the model fails to successfully critique even after $k$ times, we fall back on the critique from a stronger yet frozen model $M_C$ as the final choice. Finally, we collect dataset $D_2 = \{(Q, Att, Cri)\} \bigcup \{(Q, Att, Cri_{-1}, Att')\}$ and use $D_1 \bigcup D_2$ to train the initial model $\mathcal{M}_0$ into the final critic model $\mathcal{M}_2$, which is similar to Wang et al. (2024). This procedure helps the model to learn to criticize and refine its own reasoning outputs better.

### 3.3  INFERENCE: SELF-CHECK

To leverage our learned abilities of critique and refinement for more precise reasoning, we employ two different inference strategies: "iterative refine" and "critic as filter".

**Iterative Refine**  One single-turn refinement, which consists of multiple steps, may still contain errors. Therefore, we could iteratively inspect the refined solution, and re-refine once the critique found a mistake, and only output the final solution if it's convincing for the critic, or if it reached the maximum retry. To avoid de-generation after too many refinements, we set the maximum refine depth $d = 8$, and restart from the initial solution after $d$ unsuccessful refinement at most $n = 8$ times. Figure 3 presents a single successful round of critique and refinement.

**Critic As Filter**  Self-consistency is an effective way to reduce variance and improve accuracy. With the ability to critique, we can filter out predict-to-be-wrong answers to further boost the performance. Specifically, for the $m$ attempts $S = \{(Att, Pred)\}$, we first let our model $\mathcal{M}$ check each attempt and obtain the stepwise label, which is $S_c = \{(Att, Pred, L)\}$. And then those which detect the error at some step are filtered out and reach $S'_c = \{(Att, Pred, L) | -1 \notin L\}$. Finally, we perform the majority vote to get the answer.

## 4  EXPERIMENT

We apply the Critic-CoT training process on the dataset of GSM8K and MATH (Section 4.1), and observe a noticeable performance improvement in our trained model (Section 4.3), and out-of-domain evaluations on AGIEval and StrategyQA further exhibits the generalization of our trained critic ability (Section 4.4). We also conduct a series of ablation studies to demonstrate the effectiveness of our proposed Critic-CoT method (Section 4.5). For more analysis on the critique and refinement during test time, see Appendix A.1, and the prompt is presented in Appendix A.6.

### 4.1  SETUP

#### 4.1.1  MODEL

We fine-tune the critic-refine model on `Llama-3-70B-Instruct` (Dubey et al., 2024), which was pre-trained on more than 15 Trillion tokens and has a context length of 8,192. For critique / refinement sampling, we use `GPT4-Turbo` (OpenAI, 2023) of the version `gpt-4-0125-preview`. We use the Huggingface Transformers (Wolf et al., 2020), DeepSpeed (Rajbhandari et al., 2021) and FastChat (Zheng et al., 2023) libraries for training. We use vLLM library (Kwon et al., 2023) for model inference, adapting top-p sampling of $p = 0.95$, with temperature 0.7 for solution sampling, which follows Cobbe et al. (2021a), and 0.5 for critique and refinement. All inferences are zero-shot.

#### 4.1.2  DATASET

**Train & In-Domain Eval**  Separately, we train our model on the problem of GSM8K (Cobbe et al., 2021a) and MATH (Hendrycks et al., 2021). GSM8K is a grade-school-level math word problem

---

**Question:**
What is the smallest positive perfect cube that can be written as the sum of three consecutive integers?

**Model Attempt:**
Step 1: Let the three consecutive integers be $x - 1$, $x$, and $x + 1$. The sum of these three integers is $3x$.
Step 2: Since we are looking for a perfect cube, let's express $3x$ as a cube of some integer $n$, where $n^3 = 3x$.
Step 3: To find the smallest such $n$, we need to minimize $x$. However, since $n^3$ must be divisible by 3, and $x$ is an integer, $n$ itself must also be divisible by 3.
Step 4: The smallest integer $n$ that is divisible by 3 and gives us a perfect cube when cubed is $n = 6$, because $6^3 = 216$.
Step 5: Substituting $n = 6$ into $n^3 = 3x$, we get $216 = 3x$, which simplifies to $x = 72$.
Step 6: Therefore, the smallest positive perfect cube that can be written as the sum of three consecutive integers is $\boxed{216}$.

- - - - - - - - - - - - - - - - - - - - - - - - - - - - - - - - - - - - - - - - - - - -

**Model Critique:**
Step 1: Let the three consecutive integers be $x - 1$, $x$, and $x + 1$. The sum of these three integers is indeed $3x$.
Conclusion: Step 1 is correct.

Step 2: Expressing $3x$ as a cube of some integer $n$, where $n^3 = 3x$, correctly sets up the equation needed to solve the problem. This step accurately translates the problem statement into a mathematical expression.
Conclusion: Step 2 is correct.

Step 3: The claim that $n$ must be divisible by 3 because $n^3$ must be divisible by 3 is a misunderstanding. In reality, $n^3$ being divisible by 3 does not imply that $n$ itself must be divisible by 3; rather, it simply means that $n^3$ will always be divisible by 3 regardless of whether $n$ is divisible by 3. This step introduces an unnecessary restriction on $n$.
***Conclusion: Step 3 is incorrect.***

- - - - - - - - - - - - - - - - - - - - - - - - - - - - - - - - - - - - - - - - - - - -

**Model Refinement:**
Step 3 (Corrected): Since we are looking for the smallest positive perfect cube that can be expressed as $3x$, we should start by identifying the smallest perfect cubes and checking if any of them can be represented in this form.
Step 4: The smallest perfect cubes are $1^3 = 1, 2^3 = 8, 3^3 = 27, 4^3 = 64, 5^3 = 125, 6^3 = 216$, and so on. Among these, 27 is the first number that can be expressed as $3x$, where $x$ is an integer ($x = 9$).
Step 5: Therefore, the smallest positive perfect cube that can be written as the sum of three consecutive integers is $\boxed{27}$.

Figure 3: Criticize and refine a problem in the MATH test set. In the attempt, the model made a hallucination in Step 3, then through stepwise CoT critique, the model detected this error, and accordingly refined the solution starting from Step 3 successfully.

dataset, with 7,473 training instances and 1,319 test instances. MATH is a challenging high school math competition dataset, which consists of 7,500 training problems and 5,000 test problems. For the MATH dataset, we also follow the data split of Lightman et al. (2024), which adds 4,500 test problems into a training set and, therefore, contains 12,000 training instances and 500 representative test instances.

## 4.2 METRIC

**Solution** For the evaluation of the solution, we compute the metrics of Top-1 Accuracy `Acc` and Refine Accuracy `Refine-Acc`, in which the original Top-1 predict-answer is replaced with a refined one if the critic model found an error and made iterative refinement (Section 3.3). We also compute Majority Vote Accuracy `Maj1@N` (Wang et al., 2023b) and Majority Vote Accuracy After Critique `Critic + Maj1@N` (Section 3.3), which is to select the most frequent answer among $N$ samples, i.e. $\arg\max_a \sum_{i=1}^{N} \mathbb{1}\,(\mathbf{a}_i = a)$. Following Liu et al. (2023); Havrilla et al. (2024), we compute `Pass@N`, which select the gold answer $g$ among the $N$ predictions if present, i.e. $\arg\max_a \mathbb{1}\,(g = a)$.

**Critique** For the "evaluation of evaluation", we compute Precision, Recall and F1 for error detection. Also, we compute Critic Accuracy, where the critique should find the error in wrong answer solutions and pass the correct answer solution:

$$P = \frac{\{|Pred_i \neq Ans_i \wedge -1 \in L_i|\}}{|\{-1 \in L_i\}|}, R = \frac{\{|Pred_i \neq Ans_i \wedge -1 \in L_i|\}}{|\{Pred_i \neq Ans_i\}|}, F1 = \frac{2 * P * R}{P + R}$$

$$CriticAcc = \frac{\sum_{i=1}^{N}(Pred_i = Ans_i \wedge -1 \notin L_i) \vee (Pred_i \neq Ans_i \wedge -1 \in L_i)}{N}$$

**Out-of-Domain Eval** To further evaluate our critic model's generalization capabilities beyond mathematical tasks, we assess its performance on reasoning tasks using the StrategyQA and AGIEval datasets, which cover different domains. StrategyQA (Geva et al., 2021) is a multi-step reasoning task constructed from Wikipedia, with binary answers indicating either true or false. AGIEval (Zhong et al., 2023) comprises standardized exam questions from various fields, including college entrance exams, law school admission tests, math competitions, and lawyer qualification tests. Given the overlap with the MATH dataset, we evaluated our model using the original 7,500/5,000 training and validation split from MATH, rather than the extended 12,000/500 split.

### 4.2.1 CRITIC DATA CONSTRUCTION

**GSM8K** On GSM8K, since `GPT-4` already got 92.0% accuracy on the test set (OpenAI, 2023), which makes it hard to obtain negative data, we use `GPT-3.5-Turbo-0125` instead to sample 10 solutions for each question in the training set. Then, we use `GPT-4-Turbo` as the critic-refine model to criticize the solutions (Table 10), with $K = 16$ retry. We obtain 63,485 cases, with 49,832 positive examples and 13,653 negative examples.

In the second stage of GSM8K critique construction, we use the learned critic model to repeatedly sample until we obtain at most 5 positive and 5 negative solutions. For strong LLMs like `LLaMA-3`, it's challenging to get enough negative solutions even among 512 samples, so the size of negative data would be slightly smaller. Then, we use the learned critic model to criticize itself, also with $K = 16$ retry. In stage two, we obtain 62,877 instances, with 39,654 positive and 26,001 negative. Among the two stages, we got 126,362 instances, with 86,708 positive and 39,654 negative.

**MATH** On MATH, in the first stage, we directly use the 90,074 `GPT-4` generated solutions of PRM800K Dataset (Lightman et al., 2024), with 11,665 positive instances which all the step labels are correct, and 78,409 negative instances which one step label is incorrect. Since the MATH dataset is challenging, in order to reduce retry of `GPT-4-Turbo` and avoid not getting valid critique, for the critique of the negative solution, we additionally append reference solution in the input prompt, and hint it might contain mistakes, as suggested in prior work (Zelikman et al., 2022); for the positive solution, we simply hint it's correct. After obtaining the initial critique, we use `GPT-4-Turbo` again to remove hint phrases like "According to the reference" or "Given the hint" since we do not have any hint or reference during the test time. In stage one, we obtain 1,606 positive cases and 69,775 negative cases.

Similarly, in the second stage of MATH, we use the learned critic model to sample at most 5 positive and negative solutions. Then, we first use the critic model itself to critic its solutions, and without any hints, under $K = 16$ retry, and use `GPT-4-Turbo` to retry another $K = 16$ times with hint if failed. We construct 51,618 positive cases and 65,456 negative cases. Among the two stages, we got 188,455 cases, with 53,224 positive and 135,231 negative.

### 4.3 MAIN RESULTS

**GSM8K** The results of the GSM8K dataset highlight the effectiveness of the Critic-CoT approach in enhancing the solution accuracy. Initially, our trained model's top-1 accuracy increases from 89.6% to 91.7%, and the iterative refine strategy further enhances the accuracy to 93.3%. Additionally, the Maj1@96 method combined with the critic's filter achieves the highest accuracy of 95.4%, which is an improvement of 0.6% over the non-critic-assisted Maj1@96 approach. These results suggest that the Critic-CoT method, under relatively easy task where baseline solving accuracy is already high, can still boost performance, via critic-refine training and filtering out invalid solutions or making corrections at test time. For a concrete example of a single-turn refinement, please refer to Appendix A.5.

**MATH** As presented in Table 3, on the test set of MATH500, the baseline performance of Llama-3-70B-Instruct stands at 50.4% accuracy and Tong et al. (2024) reaches 56.1% with difficulty-aware rejection tuning, while our Critic-CoT approach initially improves the model's performance to 57.6%, with a slight increase to 57.8% through Iterative Refine. Figure 3 presents a concrete example of step-wise CoT critique, which detects an error in problem understanding in Step 3 and a successful refinement that fixes the error in Step 3 and reaches the correct answer. Compared with

| Model | Sampling Method | Acc. |
|---|---|---|
| Llama-3-70B-Instruct (Dubey et al., 2024) | - | 89.6 |
| | Maj1@96 | 94.1 |
| Llama-3.1-70B-Instruct (Dubey et al., 2024) | - | 94.5 |
| GPT4-0314 (OpenAI, 2023) | - | 92.0 |
| DeepSeek-V2 Chat-236B (DeepSeek-AI et al., 2024) | - | 92.2 |
| Qwen2-72B (Yang et al., 2024) | - | 93.2 |
| Mistral-7B: MetaMATH (Gao et al., 2024) | PRM+Maj1@256 | 87.8 |
| InternLM-MATH-20B (Ying et al., 2024) | PRM Best-of-100 | 89.3 |
| DART-Math-Llama3-70B (Tong et al., 2024) | - | 89.6 |
| DeepSeek-67B: MetaMATH (Wang et al., 2023a) | PRM+Maj1@256 | 92.5 |
| **Critic-CoT, Llama-3-70B-Instruct (*Ours*)** | - | **91.7** |
| | Iterative Refine | **93.3** ↑ 1.6 |
| | Maj1@96 | 94.8 |
| | Critic + Maj1@96 | **95.4** ↑ 0.6 |

Table 1: Solution Accuracy of GSM8K. The top-1 accuracy of our trained model improves from 89.6% to 91,7%, while iterative refinement further improves the score to 93.3%, and the critic filter increases the performance from 94.8% to 95.4%.

| Model | Sampling Method | Acc. |
|---|---|---|
| Llama-3-70B-Instruct (Dubey et al., 2024) | - | 51.0 |
| | Maj1@96 | 63.5 |
| | Maj1@512 | 64.3 |
| Llama-3.1-70B-Instruct (Dubey et al., 2024) | - | 68.0 |
| DeepSeek-V2 Chat-236B (DeepSeek-AI et al., 2024) | - | 53.9 |
| Qwen2-72B (Yang et al., 2024) | - | 69.0 |
| GPT4-0314 (OpenAI, 2023) | - | 42.5 |
| GPT4-Turbo | - | 72.6 |
| **Critic-CoT, Llama-3-70B-Instruct (*Ours*)** | - | **56.2** |
| | Iterative Refine | **56.6** ↑ 0.4 |
| | Maj1@96 | 64.2 |
| | Critic + Maj1@96 | **65.0** ↑ 0.8 |
| | Maj1@512 | 64.4 |
| | Critic + Maj1@512 | **66.4** ↑ 2.0 |

Table 2: Solution Accuracy of MATH. The top-1 accuracy of our method increases from 51.0% to 56.2%, and the effect of iterative refinement is moderate but positive improvement of 0.4%, while the performance gain of the critic filter is larger.

GSM8K, gaining from refinement is much harder. However, critic filtering still provides a notable improvement, which could be slightly easier than refinement: the accuracy rises from 64.6% with Maj1@96 to 66.6% when Critic filtering is applied, marking a 2.0% improvement. Furthermore, for Maj1@512, the accuracy rises to 68.4% after Critic filtering, showing an increase of 3.0%. While the close-source model GPT-4-MathMix achieves the highest accuracy of 78.2% with extensive sampling of 1860, the Critic-CoT approach on the open-source model can still significantly enhance the accuracy of the base model, particularly through effective error detection. The trend remains consistent with the original 7,500/5,000 split setting (Table 2). Overall, the result demonstrates the effectiveness of our method in training reasonable-level critic-refine capabilities on the challenging MATH dataset. More detailed analysis on GSM8K and MATH500 is in Appendix A.1.

### 4.4 OUT-OF-DOMAIN RESULTS

For the StrategyQA dataset, our critic models trained on two datasets show a positive performance increase when applying iterative refine and majority vote with the critic filter. On the more challenging AGIEval dataset, the critic model trained on GSM8K improves with iterative refinement, but slightly hurts the performance when filtering samples, indicating the limitations of the grade-

| Model | Sampling Method | Acc. |
|---|---|---|
| Llama-3-70B-Instruct | - | 50.4 |
| | Maj1@96 | 62.2 |
| | Maj1@512 | 63.4 |
| Mistral-7B: MetaMATH (Gao et al., 2024) | PRM+Maj1@256 | 38.6 |
| DeepSeek-67B: MetaMATH (Wang et al., 2023a) | PRM+Maj1@256 | 48.1 |
| InternLM-MATH-20B (Ying et al., 2024) | PRM Best-of-100 | 50.0 |
| DART-Math-Llama3-70B (Tong et al., 2024) | - | 56.1 |
| GPT-4-MathMix (Lightman et al., 2024) | PRM Best-of-100 | 74.5 |
| | PRM Best-of-1860 | 78.2 |
| **Critic-CoT, Llama-3-70B-Instruct (*Ours*)** | - | **57.6** |
| | Iterative Refine | **57.8** ↑ 0.2 |
| | Maj1@96 | 64.6 |
| | Critic + Maj1@96 | 66.6 ↑ 2.0 |
| | Maj1@512 | 65.4 |
| | Critic + Maj1@512 | **68.4** ↑ 3.0 |

Table 3: Solution Accuracy of MATH500. The trend is similar to the result on MATH, with a noticeable increase in top-1 accuracy and critic filtering, and a minor increase in iterative refinement.

| Model | Acc. |
|---|---|
| Llama-3-70B-Instruct | 56.6 |
| Llama-3.1-70B-Instruct | 61.8 |
| DeepSeek-V2 Chat-236B | 61.4 |
| GPT4o | 65.2 |
| **Critic-CoT, GSM8K** | 54.7 |
| - Iterative Refine | 55.6 ↑ 0.8 |
| - Maj1@96 | 60.7 |
| - Critic + Maj1@96 | 60.3 ↓ 0.4 |
| **Critic-CoT, MATH** | 59.8 |
| - Iterative Refine | **63.7** ↑ 3.9 |
| - Maj1@96 | 61.0 |
| - Critic + Maj1@96 | **61.2** ↑ 0.2 |

(a) AGIEval

| Model | Acc. |
|---|---|
| Llama-3-70B-Instruct | 76.2 |
| Llama-3.1-70B-Instruct | 84.3 |
| DeepSeek-V2 Chat-236B | 75.6 |
| GPT4-0314 | 83.6 |
| **Critic-CoT, GSM8K** | 77.5 |
| - Iterative Refine | 78.8 ↑ 1.3 |
| - Maj1@96 | 78.7 |
| - Critic + Maj1@96 | **80.5** ↑ 1.8 |
| **Critic-CoT, MATH** | 78.0 |
| - Iterative Refine | **80.1** ↑ 2.1 |
| - Maj1@96 | 78.3 |
| - Critic + Maj1@96 | 79.7 ↑ 1.4 |

(b) StrategyQA

Table 4: Solution Accuracy of AGIEval (4a) and StrategyQA (4b). Our models generally show robust generalization on out-of-domain test sets, with the exception of critic filtering by model trained on GSM8K, as the dataset is much simpler.

level critic model in handling more complex, multi-domain tasks. Conversely, the Critic-CoT model trained on MATH shows significant improvements in iterative refinement, and the method of majority vote after criticizing does not negatively impact the performance.

Overall, the results illustrate that our critic models generalize to other domains, and achieve performance improvements. This underscores the potential of our proposed critic-refine method in improving reasoning accuracy in diverse and challenging tasks beyond the training domain of math.

## 4.5 ABLATION STUDY

The results of the ablation study are shown in Table 5a and 5b, demonstrating the effectiveness of our Critic-CoT design. At the level of critique output, to assess the necessity of our proposed step-wise CoT critic, we first remove the CoT mechanism, and only train the critic model to directly predict if each step is correct (Process Label), for example, "Step 1 is correct. Step 2 is incorrect.". Then, we remove further remove the step-wise label, and let the critic model predict if the entire solution is correct, without printing anything else (Outcome Label), for example, "Some step from Step 1 to Step 4 is incorrect." or "Each step from Step 1 to Step 4 is correct.". We find that removing the Chain-of-Thought intermediate output and further step-wise labels, which fall back toward System-

| Model | Critic | | | | Refine | | Majority Vote | | |
|---|---|---|---|---|---|---|---|---|---|
| | **P** | **R** | **F1** | **Acc.** | **Init. Acc** | **Ref. Acc.** | **Pass1@N** | **Maj1@N** | **+Critic** |
| Outcome Label | **95.5** | 28.9 | 44.4 | 88.0 | 87.7 | 89.7 | 99.0 | 93.6 | 93.7 |
| Process Label | 67.9 | 22.8 | 34.1 | 89.5 | 88.0 | 89.2 | 99.0 | 93.0 | 93.0 |
| Only Refine | 30.0 | 11.4 | 16.6 | 90.8 | **92.0** | 88.2 | 98.9 | 95.2 | 95.2 |
| Only Critic | 57.1 | 31.0 | 40.2 | 91.9 | 91.2 | 91.4 | 98.9 | 94.4 | 94.5 |
| Stage 1 | 42.5 | 41.5 | 42.0 | 89.3 | 90.7 | 91.1 | 98.9 | 93.6 | 94.2 |
| Stage 2 | 50.0 | 25.0 | 33.3 | 85.5 | 90.5 | 91.3 | 99.0 | 94.4 | 94.4 |
| **Critic-CoT** | 53.3 | **58.2** | **55.7** | **92.3** | 91.7 | **93.3** | **99.1** | **94.8** | **95.4** |

(a) GSM8K

| Model | Critic | | | | Refine | | Majority Vote | | |
|---|---|---|---|---|---|---|---|---|---|
| | **P** | **R** | **F1** | **Acc.** | **Init. Acc** | **Ref. Acc.** | **Pass1@N** | **Maj1@N** | **+Critic** |
| Outcome Label | **84.4** | 39.0 | 53.3 | 63.0 | 51.8 | 53.6 | 84.0 | 56.2 | 56.2 |
| Process Label | 80.2 | 35.9 | 49.6 | 63.8 | 50.4 | 52.6 | 78.6 | 49.4 | 50.8 |
| Only Refine | 62.3 | 60.1 | 61.2 | 66.0 | 55.4 | 49.8 | **90.4** | 63.0 | 62.8 |
| Only Critic | 67.9 | 75.4 | 71.5 | 71.6 | 52.8 | 55.8 | 89.0 | 60.6 | 60.6 |
| Stage 1 | 64.6 | **93.7** | **76.5** | 69.0 | 53.2 | 41.2 | **90.4** | 63.4 | 63.0 |
| Stage 2 | 79.7 | 45.8 | 58.2 | 71.8 | 57.2 | 57.4 | **90.4** | **64.6** | 65.0 |
| **Critic-CoT** | 66.1 | 73.7 | 69.7 | **72.2** | **57.6** | **57.8** | 89.2 | **64.6** | **66.6** |

(b) MATH500

Table 5: Ablation Study on GSM8K and MATH500. We use the metrics from three aspects: critic, including precision, recall, f1-score and accuracy; Iterative Refine, including accuracy before and after the refinement; and Critic As Filter, including Pass1@96, Maj1@96, and Critic+Maj1@96. The ablation study demonstrates the effectiveness of our Critic-CoT design.

1 reasoning, negatively impacts the recall metric. Consequently, the critic model fails to detect more errors, resulting in a significantly lower critic accuracy, despite its tendency to more easily pass correct solutions.

At the training data level, to evaluate the effect of different data types, we remove the second-stage data, only use the critique and refinement produced by GPT, or remove the first-stage data and only use the critiques and refinements of self-sampled solutions. In addition, we conducted a vertical ablation by removing either the critic data or the refinement data across both stages. From the results, we find that regarding the roles of critic and refine, it is suggested that refinement contributes more to policy improvement, which echoes the finding of An et al. (2024). Yet only by combining critique and refinement during training can we enhance the policy while leveraging the critic's ability for further performance gains. Finally, training on the critique of GPT models proves better at identifying faults, but at the cost of precision. In contrast, using only the critique of itself is less effective than simply utilizing data from both stages.

## 5 CONCLUSION

In this paper, we introduced the Critic-CoT paradigm to enhance the reasoning abilities of Large Language Models, through a more System-2-like, step-by-step Chain-of-Thought critique. Our approach leverages weak supervision to construct training data for critiques and refinements, thereby reducing the reliance on extensive human annotation. We demonstrated the effectiveness of our method through substantial improvements across the dataset of GSM8K and MATH. Additionally, our results present that training on the capabilities of critique and refinement alone improves task-solving performance, which indicates a mutual-reinforce mechanism within the LLMs. We hope our work may inspire further investigations into the advancement of the self-critic framework and the transition toward System-2 reasoning.

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

# A APPENDIX

## A.1 ANALYSIS

### A.1.1 CRITIC PERFORMANCE

For both datasets, the critic model's accuracy continues to grow as the sample size $N$ increases, ultimately surpassing the performance of the majority vote, which gradually converges. Specifically, in the MATH dataset, the critic model achieves substantially higher accuracy than the solution accuracy, consistently outperforming the naive majority vote due to the critic filter's superior performance. This stark contrast highlights the critic model's effectiveness in identifying and promoting correct answers. In the GSM8K dataset, despite having a critic accuracy of only 92.3%, the critic model still manages to deliver higher accuracy gains. This outcome suggests that the critic model successfully filters answers to increase the density of correct answers and decrease the density of wrong answers, compared to the normal answer distribution. The overall results demonstrate the critic model's robust capability to enhance accuracy across different datasets, validating its practical utility in improving prediction outcomes.

### A.1.2 INSPECT ON ITERATIVE REFINE

The iterative refinement process for the GSM8K and MATH datasets demonstrates different levels of effectiveness due to their complexity, as shown in Table 6. GSM8K, being simpler, shows a higher success rate in refinement. For effective refinement, the number of false answers corrected (False

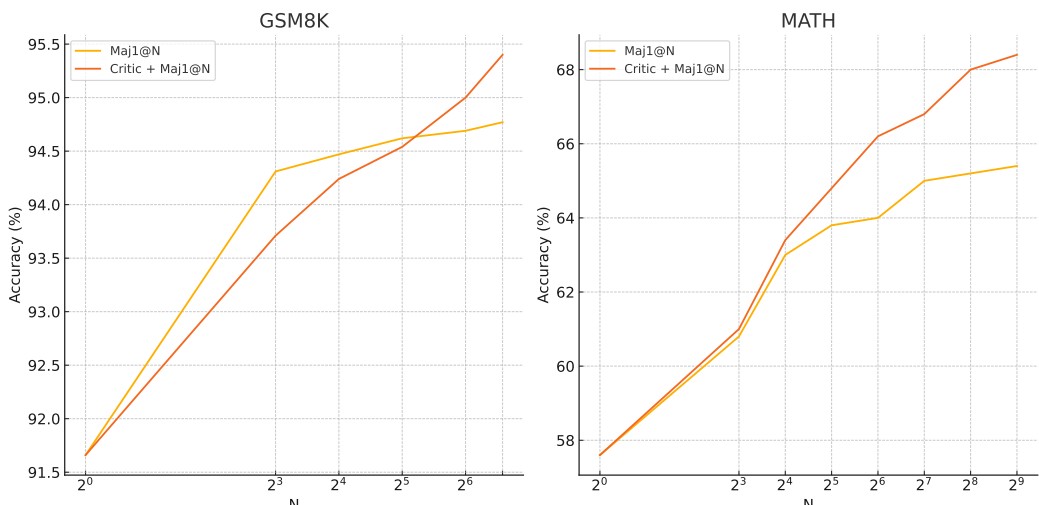

Figure 4: Performance of majority vote on GSM8K and MATH500 Datasets

| Round | Refine Acc. | True → True | False → True |
|---|---|---|---|
| 0 | 91.7 | - | - |
| 1 | 91.7 | 48.2 | 45.3 |
| 2 | 92.6 | 78.6 | 37.5 |
| 3 | 92.7 | 64.3 | **53.1** |
| 4 | 93.0 | 73.2 | 50.0 |
| 5 | 93.2 | 75.0 | **53.1** |
| 6 | 93.2 | 76.8 | **53.1** |
| 7 | **93.3** | **80.4** | 50.0 |
| 8 | **93.3** | **80.4** | 50.0 |

(a) GSM8K

| Round | Refine Acc. | True → True | False → True |
|---|---|---|---|
| 0 | 57.6 | - | - |
| 1 | 53.4 | 29.0 | 17.7 |
| 2 | 57.2 | **65.7** | 13.9 |
| 3 | 55.2 | 48.6 | **15.2** |
| 4 | 57.2 | 60.9 | 15.9 |
| 5 | 57.4 | 60.0 | 17.1 |
| 6 | 57.6 | 61.4 | 17.1 |
| 7 | **57.8** | 60.0 | **18.4** |
| 8 | **57.8** | 62.9 | 16.5 |

(b) MATH500

Table 6: Iterative Refine on GSM8K (6a) and MATH500 (6b).

→ True) must exceed the number of true answers incorrectly changed (True → False). Despite occasional mistakes by the critic, correct answers are not always altered incorrectly.

For GSM8K (Table 6a), accuracy improves from 91.7% initially to 93.3% by the seventh round, with significant gains in both true-to-true and false-to-true transformations. In contrast, MATH (Table 6b) starts at 57.6% accuracy, reaching 57.8% by the seventh round. The iterative refinement process tends to converge, which is expected.

### A.1.3 GROUP BY DIFFICULTY LEVEL

For the MATH dataset, the difficulty level is given from 1 to 5. For the GSM8K dataset, we set the difficulty level according to the number of expressions $n$ that appeared in the reference solution, i.e., $max(1, min(5, n))$. As illustrated in Figure 5, the performance on the GSM8K dataset shows a gradual decline as the difficulty level increases. This trend is accompanied by the emerging effects of the critic and refine stages, which become more prominent at higher difficulty levels. In contrast, the accuracy on the MATH dataset declines sharply as the problems become more challenging. Generally, the refine stage proves effective across all levels, while the critic stage is beneficial at most levels, with some minor exceptions. These observations suggest potential areas for further improvements in the critic mechanism.

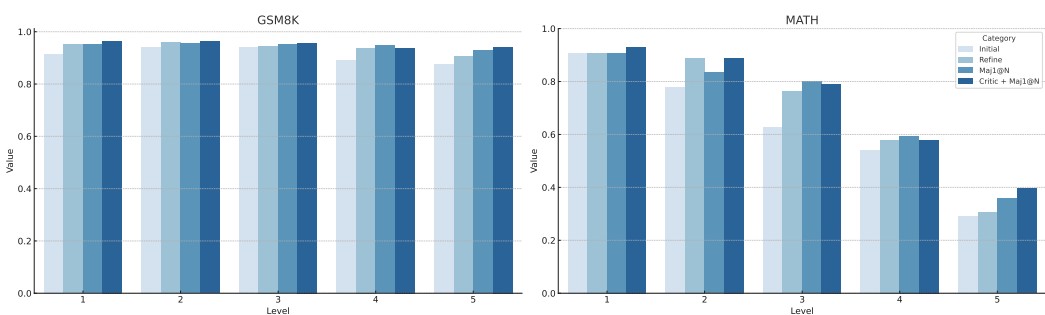

Figure 5: Performance group by difficulty level, on GSM8K and MATH500 Datasets

## A.2 DISCUSSION

### A.2.1 DIFFERENCES BETWEEN CRITIC-CoT AND REFLEXION

We adopt a similar approach to Relexion (Shinn et al., 2023), which leverages natural language critique to facilitate refinement, but our method diverges in the following ways:

**Step-wise CoT Critique** Reflexion translates and augments the binary reward signal from the environment to natural language, but on an instance level. Instead, fine-grained Chain-of-Thought analysis at the step level, which is more systematic, and enables us to locate the error and start refinement from a specific step, rather than refine the whole attempt.

**Enhanced Critic ability** While Relexion proposed an in-context learning pipeline for policy optimization under the oracle success/fail binary feedback signal, Huang et al. (2024) showed that without external feedback, vanilla LLMs cannot self-correct effectively due to limited critique ability. Therefore, to teach the LLMs the ability of intrinsic self-critique, our approach tries to learn the critique ability itself, through Critic-CoT training, and can apply it to test-time situations where the oracle feedback signal is not available.

### A.2.2 SOURCE OF IMPROVEMENT

As discussed in Sections 4.3 and 4.5, the improvements can be attributed to two key factors: training with critique and refinement, and application of critique and refinement at test time.

- Strengthening the ability to critique and refinement would not compromise the task-solving performance, but improve it. Therefore, our critic model already exhibits noticeable performance improvements in task-solving, even in the absence of additional critique steps during the inference phase.
- At the inference phase, we can actively leverage the model's ability to reflect on its reasoning and correct mistakes via Critic As Filter and Iterative Refinement, which leads to additional improvements.

### A.2.3 COMPARISON BETWEEN INFERENCE METHODS

As the results in Table 1, 2, 3 shows, the performance of majority vote and Critic As Filter surpass Iterative Refinement. We believe it's due to the intrinsic challenges of refinement and the relatively limited search space.

On the one hand, for Iterative Refinement to work properly, it requires the model to 1) Detect errors on an attempt; 2) Refine the mistakes; 3) Exit if no further errors are detected. This pipeline could be more sensitive to error accumulation. Moreover, it only edits on a single example and has a limited retry, which is sample-efficient, but may not explore the solution space more actively, as majority vote does. Specifically on the dataset of GSM8K, the invocation statistics are as follows:

- Majority vote: 1319 * 96 = 126,624

- Iterative Refinement: among 1319 test cases, our Critic-CoT model predicts 274 problematic instances and iterates 1627 times (on average 5.94 rounds for each wrong case), which makes in total 1319 * 2 + 1627 * 2 = 5892 invocations, which is 21.5 times fewer than Majority Vote calls.

On the other hand, majority vote is a strong baseline, as it requires massive sampling. It leverages diverse reasoning paths and tries to mitigate the stochastic of a single sample. But under the method of Critic As Filter, we actively filter out problematic attempts and perform the majority vote on the more reasonable candidates, rather than equally account for all the predicted answers as the vanilla majority vote does, which further increases performance. This in turn demonstrates our model's strong ability to critique.

### A.2.4 Self-Reflection

Besides the main results, through out-of-domain evaluation in Table 4, we find our model demonstrates generalized ability to critique and refine. While the ability of LLMs to self-reflect still remains an open question, and we hope our work as a valuable exploration could shed light on future studies in this area.

Moreover, as long as we adequately improve the models' ability to critique, we could achieve test-time performance increase in the form of "self-reflection". As the experiment results present, after Critic-CoT training, the ability to critique and generate both improves, though they are not exactly identical. Notably, the critique ability can surpass the task-solving ability, allowing the model to detect errors even when it has a low probability of generating a valid solution, as prior works (Saunders et al., 2022; Lin et al., 2024) also suggest. This indicates that by strengthening the model's CoT critique ability beyond its generation capability, we can leverage this discriminative power to reject imperfect responses and achieve positive performance gains.

### A.3 Manual Evaluation on Constructed Data

**Process Correctness of Correct Answer Attempt** We each sample 100 correct answer solutions, on GSM8K by GPT-3.5-Turbo and GPT-4-Turbo, and MATH by GPT-4-Turbo, and manually check if all intermediate steps are correct. The results are demonstrated in Table 7. We find that in general, the correct final answer is a good indicator of correct intermediate steps. Also, from GSM8K to MATH, as the reasoning traces become longer and more complicated, the percentage of correct answer but with wrong intermediate steps increases.

| Data | Model | Intermediate Accuracy of Correct Answer Attempt |
|------|-------|------------------------------------------------|
| GSM8K | GPT-3.5-Turbo | 97% |
| GSM8K | GPT-4-Turbo | 99% |
| MATH | GPT-4-Turbo | 93% |

Table 7: Human Evaluation on the Intermediate Steps of Attempts with Correct Answer

**Quality of Critique and Refinement in Training Data** We sampled 100 entries (50 with the correct answer and 50 with the wrong answer) each from the critic-cot data on GSM8K and MATH, and conducted a manual verification to verify the accuracy of the step-wise critiques. For the critique of the correct answer attempt, it is valid if there is indeed no error in all the intermediate steps; for the critique of the wrong attempt, it is valid if the first error step and the reason for the error are both identified. A refinement is correct, if the continuation steps are flawless.

The results of manual verification are demonstrated in Table 8, and we find that the data we automatically constructed maintains a high level of accuracy at the step level, which can well support the critique training process.

### A.4 Answer Extraction

We let the model print the answer in the format `\boxed{answer}`. The model generates the answer following this pattern. We then extract the regular expression `\boxed{.*}` from the model

| Data | Critique of Wrong Attempt | Refinement of Wrong Attempt | Critique of Correct Answer Attempt |
|------|---------------------------|------------------------------|-------------------------------------|
| GSM8K | 86% | 96% | 100% |
| MATH | 84% | 96% | 94% |

Table 8: Human Evaluation on the critique and refinement of Critic-CoT Training Data

output, and obtain the valid answer expression with matched parenthesis. The Python code for answer extraction is shown in Table 9.

```python
import re

def extract_boxed_expressions_custom(text):
    stack = []
    current_expr = ""
    i = 0
    while i < len(text):
        if text[i:i+7] == r"\boxed{":
            if stack:
                current_expr += text[i]
            stack.append("{")
            i += 7
        elif text[i] == "{" and stack:
            stack.append("{")
            current_expr += text[i]
            i += 1
        elif text[i] == "}" and stack:
            stack.pop()
            if stack:
                current_expr += text[i]
            else:
                current_expr = re.split('=', current_expr)[-1]
                return current_expr
            i += 1
        elif stack:
            current_expr += text[i]
            i += 1
        else:
            i += 1
    return ''

def extract_answer(text):
    predict_answer_span = re.findall(r'(\boxed{.*})', text, re.DOTALL)[-1]
    predict_answer = extract_boxed_expressions_custom(predict_answer_span)
    return predict_answer
```

Table 9: Python code for extracting the boxed answer.

## A.5 AN EXAMPLE OF REFINEMENT ON GSM8K

As presented in Figure 6, the model forgot to add one year at Step 3; then, through CoT critique, the model found that while Step 1 and Step 2 are correct, Step 3 contains this ignorance error. Finally, guided by the critique of Step 3, the model made a correction and reach the gold answer of 13.

## A.6 PROMPTS

Table 10, Table 13, and Table 11 presents the prompt for critic-refine data collection using GPT4-Turbo, with Table 12 for removing the hint phrases (Section 3.2). Table 14, Table 15, and Table 16 shows the prompt of trained model for solving, critique, and refinement during stage-2-training (Section 3.2) and inference (Section 3.3). Table 17, Table 18 and Table 19 present the prompts and responses of a single turn critique-refinement, under Critic-CoT, Step-wise Label Critic and Final Label Critic respectively.

---

**Question:**
Carlos is planting a lemon tree. The tree will cost $90 to plant. Each year it will grow 7 lemons, which he can sell for $1.5 each. It costs $3 a year to water and feed the tree. How many years will it take before he starts earning money on the lemon tree?

**Model Attempt:**
Step 1: Calculate the annual income from selling lemons. This is done by multiplying the number of lemons grown per year (7) by the price per lemon ($1.5). So, 7 * $1.5 = $10.50.
Step 2: Subtract the annual cost of watering and feeding the tree ($3) from the annual income from selling lemons ($10.50). So, $10.50 - $3 = $7.50. This is the net income per year after expenses.
Step 3: Determine when Carlos will start earning money back. This is done by dividing the initial cost of planting the tree ($90) by the net income per year ($7.50). So, $90 / $7.50 = 12 years.

Therefore, it will take Carlos $\boxed{12}$ years before he starts earning money on the lemon tree.

- - - - - - - - - - - - - - - - - - - - - - - - - - - - - - - - - - - - - - - - - - -

**Model Critique:**
Step 1: The attempt calculates the annual income from selling lemons by multiplying the number of lemons grown per year (7) by the price per lemon ($1.5), resulting in $10.50.
Conclusion: Step 1 is correct.

Step 2: The attempt subtracts the annual cost of watering and feeding the tree ($3) from the annual income from selling lemons ($10.50), resulting in a net income per year of $7.50.
Conclusion: Step 2 is correct.

Step 3: The attempt divides the initial cost of planting the tree ($90) by the net income per year ($7.50), resulting in 12 years. However, this calculation does not account for the fact that Carlos will start earning money back in the first year, even though he won't break even until the 13th year.
***Conclusion: Step 3 is incorrect.***

- - - - - - - - - - - - - - - - - - - - - - - - - - - - - - - - - - - - - - - - - - -

**Model Refinement:**
Step 3 (Corrected): Determine when Carlos will start earning money back. This is done by dividing the initial cost of planting the tree ($90) by the net income per year ($7.50). So, $90 / $7.50 = 12 years. However, since Carlos will start earning money back in the first year, we need to add one more year to this calculation. Therefore, it will take Carlos $\boxed{13}$ years before he starts earning money on the lemon tree.

Figure 6: Criticize and refine a problem in the GSM8K test set.

---

**Prompt**

---

How do you evaluate the following attempt with respect to the problem?

```
<problem>
```
{problem}
```
</problem>
```

```
<attempt>
```
{attempt}
```
</attempt>
```

```
-----
```
**Notes**:
- Please think step by step.
- Your reasoning should precede any claims or conclusions you make to avoid unwarranted assertions.
- At the end of the evaluation for each step, YOU MUST articulate the conclusion using the format "Conclusion: Step [i] is correct" or "Conclusion: Step [i] is incorrect". Words like "partially correct" are prohibited.
- You shall not evaluate multiple steps at a time, so words like "Step 7 to Step 24:" or "Step 4 through 6" are forbidden.
- Once a mistake is identified and stated, stop the evaluation, and enumerate the corrected steps starting from the step where the mistake was detected, and label this part of your response with `<correction>` at the start and `</correction>` at the end. Also, the final answer should be a single number, in the form \boxed{}, at the final step.

---

Table 10: The prompt for the collection of critique and refinement on GSM8K, using GPT4-Turbo.

---

**Prompt**

---

How do you evaluate the following attempt with respect to the problem, with the help of reference solution?
Hint: There could be a mistake.

```
<problem>
{problem}
</problem>

<reference_solution>
{reference_solution}
</reference_solution>

<attempt>
{attempt}
</attempt>

-----
```
**Notes**:
- Please think step by step.
- Your reasoning should precede any claims or conclusions you make to avoid unwarranted assertions.
- Please ensure that the output text does not include phrases implying the use of a reference solution or hint, even though these resources are being utilized.
- At the end of the evaluation for each step, YOU MUST articulate the conclusion using the format "Conclusion: Step [i] is correct" or "Conclusion: Step [i] is incorrect". Words like "partially correct" are prohibited.
- You shall not evaluate multiple steps at a time, so words like "Step 7 to Step 24:" or "Step 4 through 6" are forbidden.
- Once a mistake is identified and stated, stop the evaluation, and enumerate the corrected steps starting from the step where the mistake was detected, and label this part of your response with `<correction>` at the start and `</correction>` at the end. Also, the final answer should be in the form \boxed{}, at the final step.

---

Table 11: The prompt for the collection of critique and refinement on MATH incorrect attempt, using GPT4-Turbo.

---

**Prompt**

---

For the following text, remove any phrases like "reference solution" or "hint", and keep all the other content. Do not miss the "`<correction>`" and "`</correction>`" labels that exist in the text. Do not respond to anything else.

```
-----
```
{critique_refinement}

---

Table 12: The prompt for removing the hint of critique and refinement on MATH, using GPT4-Turbo.

---

**Prompt**

---

How do you evaluate the following attempt with respect to the problem?
Hint: All the steps are correct, and the attempt reached a correct answer.

```
<problem>
```
{problem}
```
</problem>
```

```
<attempt>
```
{attempt}
```
</attempt>
```

```
-----
```
**Notes**:
- Please think step by step.
- Your reasoning should precede any claims or conclusions you make to avoid unwarranted assertions.
- Please ensure that the output text does not include phrases implying the use of a reference solution or hint, even though these resources are being utilized.
- At the end of the evaluation for each step, YOU MUST articulate the conclusion using the format "Conclusion: Step [i] is correct" or "Conclusion: Step [i] is incorrect". Words like "partially correct" are prohibited.
- You shall not evaluate multiple steps at a time, so words like "Step 7 to Step 24:" or "Step 4 through 6" are forbidden.
- Once a mistake is identified and stated, stop the evaluation, and enumerate the corrected steps starting from the step where the mistake was detected, and label this part of your response with `<correction>` at the start and `</correction>` at the end. Also, the final answer should be in the form \boxed{}, at the final step.

---

Table 13: The prompt for the collection of critique on MATH correct attempt, using GPT4-Turbo.

---

**Prompt**

---

## Problem
{problem}

```
-----
```
Solve the problem step by step, marking each step as "Step [i]:".
Your final answer should be in the form \boxed{answer}, at the end of your response.

---

Table 14: The prompt for solution sampling.

**Prompt**

How do you evaluate the following attempt with respect to the problem?

```
<problem>
{problem}
</problem>

<attempt>
{attempt}
</attempt>

-----
```
**Notes**:
- Please think step by step.
- Your reasoning should precede any claims or conclusions you make to avoid unwarranted assertions.
- At the end of the evaluation for each step, YOU MUST articulate the conclusion using the format "Conclusion: Step [i] is correct" or "Conclusion: Step [i] is incorrect". Words like "partially correct" are prohibited.

Table 15: The prompt for critique.

**Prompt**

How do you refine the following attempt with respect to the problem, given the criticism?

```
<problem>
{problem}
</problem>

<attempt>
{attempt}
</attempt>

<criticism>
{wrong_step_criticism}
</criticism>
```

Table 16: The prompt for refinement.

**Prompt of Critique under Critic-CoT**

How do you evaluate the following attempt with respect to the problem?

<problem>
Consider the geometric sequence $\frac{125}{9}, \frac{25}{3}, 5, 3, \ldots$. What is the eighth term of the sequence? Express your answer as a common fraction.
</problem>

<attempt>
Step 1: Identify the first term of the sequence.
The first term $a_1$ is:
$$a_1 = \frac{125}{9}$$

Step 2: Calculate the common ratio $r$.
The common ratio $r$ can be found by dividing the second term $a_2$ by the first term $a_1$:
$$r = \frac{a_2}{a_1} = \frac{\frac{25}{3}}{\frac{125}{9}} = \frac{25}{3} \times \frac{9}{125}$$

Calculating the right-hand side:
$$= \frac{25 \cdot 9}{3 \cdot 125} = \frac{225}{375}$$

Now simplify $\frac{225}{375}$:
$$= \frac{225 \div 75}{375 \div 75} = \frac{3}{5}$$

Thus, the common ratio is:
$$r = \frac{3}{5}$$

Step 3: Use the formula for the $n$-th term of a geometric sequence.
The $n$-th term of a geometric sequence can be calculated using the formula:
$$a_n = a_1 \cdot r^{n-1}$$

Step 4: Calculate the eighth term $a_8$.
Substituting $n = 8$ into the formula:
$$a_8 = a_1 \cdot r^{8-1} = \frac{125}{9} \cdot \left(\frac{3}{5}\right)^7$$

Step 5: Calculate $\left(\frac{3}{5}\right)^7$.
Calculating $\left(\frac{3}{5}\right)^7$:
$$\left(\frac{3}{5}\right)^7 = \frac{3^7}{5^7} = \frac{2187}{78125}$$

Step 6: Substitute back into the expression for $a_8$.
Now we compute:
$$a_8 = \frac{125}{9} \cdot \frac{2187}{78125}$$

Step 7: Multiply the fractions.
To multiply:
$$a_8 = \frac{125 \cdot 2187}{9 \cdot 78125}$$

Calculating the denominator:
$$9 \cdot 78125 = 703125$$

---

**Prompt of Critique under Critic-CoT**

Calculating the numerator:
$$125 \cdot 2187 = 273375$$

Thus:
$$a_8 = \frac{273375}{703125}$$

Step 8: Simplify the fraction $\frac{273375}{703125}$.

To simplify: First, calculate the GCD of the numerator and denominator. We can find that both 273375 and 703125 are divisible by 135.

Calculating:
$$\frac{273375 \div 135}{703125 \div 135} = \frac{2025}{5205}$$

Continuing to simplify, both 2025 and 5205 are divisible by 15:
$$\frac{2025 \div 15}{5205 \div 15} = \frac{135}{347}$$

This is now fully simplified because 135 and 347 have no common factors.

Final Answer:

Thus, the eighth term of the sequence is:

$$\boxed{\frac{135}{347}}$$

```
</attempt>
```

```
-----
```
**Notes**:
- Please think step by step.
- Your reasoning should precede any claims or conclusions you make to avoid unwarranted assertions.
- At the end of the evaluation for each step, YOU MUST articulate the conclusion using the format "Conclusion: Step [i] is correct" or "Conclusion: Step [i] is incorrect". Words like "partially correct" are prohibited.

**Response of Critique under Critic-CoT**

Step 1: The first term $a_1 = \frac{125}{9}$ is identified correctly.

Conclusion: Step 1 is correct.

Step 2: The common ratio $r$ is calculated as:

$$r = \frac{a_2}{a_1} = \frac{\frac{25}{3}}{\frac{125}{9}} = \frac{25}{3} \times \frac{9}{125} = \frac{225}{375} = \frac{3}{5}$$

This calculation is correct.

Conclusion: Step 2 is correct.

Step 3: The formula for the $n$-th term is correctly stated as:

$$a_n = a_1 \cdot r^{n-1}$$

Conclusion: Step 3 is correct.

Step 4: The eighth term is computed as:

$$a_8 = \frac{125}{9} \cdot \left(\frac{3}{5}\right)^7$$

This is correct.

Conclusion: Step 4 is correct.

---

**Response of Critique under Critic-CoT**

Step 5: Calculating $\left(\frac{3}{5}\right)^7$:

$$\left(\frac{3}{5}\right)^7 = \frac{2187}{78125}$$

This calculation is correct.
Conclusion: Step 5 is correct.
Step 6: Substituting back:

$$a_8 = \frac{125}{9} \cdot \frac{2187}{78125}$$

This is correct.
Conclusion: Step 6 is correct.
Step 7: Multiplying the fractions:

$$a_8 = \frac{125 \cdot 2187}{9 \cdot 78125}$$

The denominator calculation is correct:

$$9 \cdot 78125 = 703125$$

However, for the numerator:

$$125 \cdot 2187 = 273375$$

This is also correct.
Conclusion: Step 7 is correct.
Step 8: The goal is to simplify the fraction $\frac{273375}{703125}$. The attempt made to find the GCD and simplify. To simplify the fraction, we need to find the greatest common divisor (GCD) of 273375 and 703125.
The standard approach involves either prime factorization or using the Euclidean algorithm.
Prime Factorization:
- 273375:
- Dividing by 5: $273375 \div 5 = 54675$
- Dividing by 5 again: $54675 \div 5 = 10935$
- Dividing by 5 again: $10935 \div 5 = 2187$
- The number 2187 is $3^7$.
- Therefore, $273375 = 5^3 \cdot 3^7$.

- 703125:
- Dividing by 5: $703125 \div 5 = 140625$
- Dividing by 5 again: $140625 \div 5 = 28125$
- Dividing by 5 again: $28125 \div 5 = 5625$
- Dividing by 5 again: $5625 \div 5 = 1125$
- Dividing by 5 again: $1125 \div 5 = 225$
- Dividing by 5 again: $225 \div 5 = 45$
- Finally, $45 = 3^2 \cdot 5^1$.
- Therefore, $703125 = 5^6 \cdot 3^2$.

The GCD can now be derived from the lowest powers of the common factors:
- For 5: $5^3$
- For 3: $3^2$
Thus, the GCD is:

$$5^3 \cdot 3^2$$

Therefore, 273375 and 703125 are not divisible by 135.
Conclusion: Step 8 is incorrect.

---

**Prompt of Refinement under Critic-CoT**

---

How do you refine the following attempt with respect to the problem, given the criticism?

```
<problem>
```
Consider the geometric sequence $\frac{125}{9}, \frac{25}{3}, 5, 3, \ldots$. What is the eighth term of the sequence? Express your answer as a common fraction.
```
</problem>
```

```
<attempt>
```
Step 1: Identify the first term of the sequence.
The first term $a_1$ is:
$$a_1 = \frac{125}{9}$$

Step 2: Calculate the common ratio $r$.
The common ratio $r$ can be found by dividing the second term $a_2$ by the first term $a_1$:
$$r = \frac{a_2}{a_1} = \frac{\frac{25}{3}}{\frac{125}{9}} = \frac{25}{3} \times \frac{9}{125}$$

Calculating the right-hand side:
$$= \frac{25 \cdot 9}{3 \cdot 125} = \frac{225}{375}$$

Now simplify $\frac{225}{375}$:
$$= \frac{225 \div 75}{375 \div 75} = \frac{3}{5}$$

Thus, the common ratio is:
$$r = \frac{3}{5}$$

Step 3: Use the formula for the $n$-th term of a geometric sequence.
The $n$-th term of a geometric sequence can be calculated using the formula:
$$a_n = a_1 \cdot r^{n-1}$$

Step 4: Calculate the eighth term $a_8$.
Substituting $n = 8$ into the formula:
$$a_8 = a_1 \cdot r^{8-1} = \frac{125}{9} \cdot \left(\frac{3}{5}\right)^7$$

Step 5: Calculate $\left(\frac{3}{5}\right)^7$.
Calculating $\left(\frac{3}{5}\right)^7$:
$$\left(\frac{3}{5}\right)^7 = \frac{3^7}{5^7} = \frac{2187}{78125}$$

Step 6: Substitute back into the expression for $a_8$.
Now we compute:
$$a_8 = \frac{125}{9} \cdot \frac{2187}{78125}$$

Step 7: Multiply the fractions.
To multiply:
$$a_8 = \frac{125 \cdot 2187}{9 \cdot 78125}$$

Calculating the denominator:
$$9 \cdot 78125 = 703125$$

Calculating the numerator:

---

---

**Prompt of Refinement under Critic-CoT**

---

$$125 \cdot 2187 = 273375$$

Thus:

$$a_8 = \frac{273375}{703125}$$

Step 8: Simplify the fraction $\frac{273375}{703125}$.

To simplify: First, calculate the GCD of the numerator and denominator. We can find that both 273375 and 703125 are divisible by 135.

Calculating:

$$\frac{273375 \div 135}{703125 \div 135} = \frac{2025}{5205}$$

Continuing to simplify, both 2025 and 5205 are divisible by 15:

$$\frac{2025 \div 15}{5205 \div 15} = \frac{135}{347}$$

This is now fully simplified because 135 and 347 have no common factors.

Final Answer:

Thus, the eighth term of the sequence is:

$$\boxed{\frac{135}{347}}$$

```
</attempt>
```

```
<criticism>
```
Step 8: The goal is to simplify the fraction $\frac{273375}{703125}$. The attempt made to find the GCD and simplify. To simplify the fraction, we need to find the greatest common divisor (GCD) of 273375 and 703125.

The standard approach involves either prime factorization or using the Euclidean algorithm.

Prime Factorization:

- 273375:
- Dividing by 5: $273375 \div 5 = 54675$
- Dividing by 5 again: $54675 \div 5 = 10935$
- Dividing by 5 again: $10935 \div 5 = 2187$
- The number 2187 is $3^7$.
- Therefore, $273375 = 5^3 \cdot 3^7$.

- 703125:
- Dividing by 5: $703125 \div 5 = 140625$
- Dividing by 5 again: $140625 \div 5 = 28125$
- Dividing by 5 again: $28125 \div 5 = 5625$
- Dividing by 5 again: $5625 \div 5 = 1125$
- Dividing by 5 again: $1125 \div 5 = 225$
- Dividing by 5 again: $225 \div 5 = 45$
- Finally, $45 = 3^2 \cdot 5^1$.
- Therefore, $703125 = 5^6 \cdot 3^2$.

The GCD can now be derived from the lowest powers of the common factors:
- For 5: $5^3$
- For 3: $3^2$
Thus, the GCD is:

$$5^3 \cdot 3^2$$

Therefore, 273375 and 703125 are not divisible by 135.

Conclusion: Step 8 is incorrect.

```
</criticism>
```

---

**Prompt of Refinement under Critic-CoT**

---

```
<correction>
```
Step 8: Simplify $\frac{273375}{703125}$. First, we divide it by 375.
Calculating:

$$\frac{273375 \div 375}{703125 \div 375} = \frac{729}{1875}$$

Step 9: Further simplify $\frac{729}{1875}$. The numerator 729 is $3^6$ and the denominator 1875 is $3 \cdot 625$. The GCD is 3.
Calculating:

$$\frac{729 \div 3}{1875 \div 3} = \frac{243}{625}$$

This fraction is fully simplified.
Final Answer: Thus, the eighth term of the sequence is:

$$\boxed{\frac{243}{625}}$$

```
</correction>
```

---

Table 17: Full example of Critic-CoT. The model got the wrong common factor in Step 8, and through CoT critique, identified this mistake, then took the refinement and reached the correct answer.

---

**Prompt of Critique under Step-wise Label Critic**

---

How do you evaluate the following attempt with respect to the problem?

```
<problem>
[...]
</problem>

<attempt>
[...]
</attempt>

-----
```
**Notes**:
- For each step, you only need to output "Step [i] is correct" if right, or "Step [i] is incorrect" if wrong. Do not provide anything else in your output.

**Response of Critique under Step-wise Label Critic**

---

Step 1 is correct.
Step 2 is correct.
Step 3 is correct.
Step 4 is correct.
Step 5 is incorrect.

---

1620
1621
1622
1623
1624
1625
1626
1627
1628
1629
1630
1631
1632
1633
1634
1635
1636
1637
1638
1639
1640
1641
1642
1643
1644
1645
1646
1647
1648
1649
1650
1651
1652
1653
1654
1655
1656
1657
1658
1659
1660
1661
1662
1663
1664
1665
1666
1667
1668
1669
1670
1671
1672
1673

**Prompt of Refinement under Step-wise Label Critic**

How do you refine the following attempt with respect to the problem, given the criticism?

```
<problem>
[...]
</problem>

<attempt>
[...]
</attempt>

<criticism>
```
Step 5 is incorrect.
```
</criticism>
```

**Response of Refinement under Step-wise Label Critic**

```
<correction>
```
Step 5: Calculate $\left(\frac{3}{5}\right)^7$.
$$\left(\frac{3}{5}\right)^7 = \frac{3^7}{5^7} = \frac{2187}{78125} \text{ (correct)}$$

Step 6: Substitute back into the expression for $a_8$.
$$a_8 = \frac{125}{9} \cdot \frac{2187}{78125}$$

Step 7: Multiply the fractions. Calculating the numerator:
$$125 \cdot 2187 = 273375 \text{ (correct)}$$

Calculating the denominator:
$$9 \cdot 78125 = 703125 \text{ (correct)}$$

Step 8: Simplify the fraction $\frac{273375}{703125}$. To simplify: Calculating the GCD shows that both $273375$ and $703125$ are divisible by $675$, so:
$$\frac{273375 \div 675}{703125 \div 675} = \frac{405}{1045}$$

Continuing to simplify, both 405 and 1045 are divisible by 5:
$$\frac{405 \div 5}{1045 \div 5} = \frac{81}{209}$$

Thus, the final answer is:
$$\boxed{\frac{81}{209}}$$

```
</correction>
```

Table 18: Full example of Step-wise Label Critic. The problem and the attempt are the same in Table 17, so we skip them. Without CoT critique, the model incorrectly believed that Step 5 is wrong, then made the refinement starting from Step 5, which is unsuccessful.

---

**Prompt of Critique under Outcome Label Critic**

How do you evaluate the following attempt with respect to the problem?

```
<problem>
[...]
</problem>

<attempt>
[...]
</attempt>

-----
```
**Notes**:
- For the whole attempt, you only need to output "Each step from Step 1 to Step 8 is correct." if right, or "Some step from Step 1 to Step 8 is incorrect" if wrong. Do not provide anything else in your output.

---

**Response of Critique under Outcome Label Critic**

Some step from Step 1 to Step 8 is incorrect.

---

**Prompt of Refinement under Outcome Label Critic**

How do you refine the following attempt with respect to the problem, given the criticism?

```
<problem>
[...]
</problem>

<attempt>
[...]
</attempt>

<criticism>
```
Some step from Step 1 to Step 8 is incorrect.
```
</criticism>
```

---

**Response of Refinement under Outcome Label Critic**

```
<correction>
```
Step 1: Identify the first term of the sequence.
The first term $a_1$ is:

$$a_1 = \frac{125}{9}$$

Step 2: Calculate the common ratio $r$.
The common ratio $r$ can be found by dividing the second term $a_2$ by the first term $a_1$:

$$r = \frac{a_2}{a_1} = \frac{\frac{25}{3}}{\frac{125}{9}} = \frac{25}{3} \times \frac{9}{125} = \frac{225}{375}$$

Now simplify $\frac{225}{375}$:

$$= \frac{225 \div 75}{375 \div 75} = \frac{3}{5}$$

Thus, the common ratio is:

$$r = \frac{3}{5}$$

Step 3: Use the formula for the $n$-th term of a geometric sequence.

---

---

**Response of Refinement under Outcome Label Critic**

---

The $n$-th term of a geometric sequence can be calculated using the formula:

$$a_n = a_1 \cdot r^{n-1}$$

Step 4: Calculate the eighth term $a_8$.
Substituting $n = 8$ into the formula:

$$a_8 = a_1 \cdot r^{8-1} = \frac{125}{9} \cdot \left(\frac{3}{5}\right)^7$$

Step 5: Calculate $\left(\frac{3}{5}\right)^7$.
Calculating $\left(\frac{3}{5}\right)^7$:

$$\left(\frac{3}{5}\right)^7 = \frac{3^7}{5^7} = \frac{2187}{78125}$$

Step 6: Substitute back into the expression for $a_8$.
Now we compute:

$$a_8 = \frac{125}{9} \cdot \frac{2187}{78125}$$

Step 7: Multiply the fractions.
To multiply:

$$a_8 = \frac{125 \cdot 2187}{9 \cdot 78125}$$

Calculating the denominator:

$$9 \cdot 78125 = 703125$$

Calculating the numerator:

$$125 \cdot 2187 = 273375$$

Thus:

$$a_8 = \frac{273375}{703125}$$

Step 8: Simplify the fraction $\frac{273375}{703125}$.
To simplify: Finding the GCD, both 273375 and 703125 are divisible by 675:

$$\frac{273375 \div 675}{703125 \div 675} = \frac{405}{1042}$$

Now we check if $\frac{405}{1042}$ can be simplified further. Both numbers do not have common factors.
Final Answer:
Thus, the eighth term of the sequence is:

$$\boxed{\frac{405}{1042}}$$

`</correction>`

---

Table 19: Full example of Final Label Critic. The problem and the attempt are also the same in Table 17. With only the outcome label, the model managed to detect the attempt was incorrect, then made the refinement starting from the first step, which was unsuccessful.

