# OpenReview forum: "Critic-CoT: Boosting the reasoning abilities of large language model via Chain-of-Thought Critic"
_ICLR.cc/2025/Conference — Submitted to ICLR 2025_

### Official Review · Reviewer_GCMB · 2024-11-03

**Soundness:** 3
**Presentation:** 2
**Contribution:** 3
**Rating:** 8
**Confidence:** 4

**Summary:**

The paper addresses the limitations of existing self-critique methods for LLMs, which primarily use simple feedback mechanisms that resemble intuitive (System-1) processing. In fact, many works have explored this issue, such as whether large language models (LLMs) can accurately reflect and how to inspire LLMs to reflect. Most works are designed based on prompt engineering.

To bridge this gap, the authors propose Critic-CoT, a framework designed to elevate LLMs' reasoning by developing a stepwise, chain-of-thought (CoT) critique process. The motivation is to enable LLMs to critique and refine their solutions autonomously, thus enhancing task-solving abilities without human annotation costs.  The author generates step-wise feedback by extensive sampling and using a strong LLM. Then based on this self-generated feedback dataset, it trains the model to have the ability of self-criticism.

**Strengths:**

First, the motivation of this paper is very interesting. How can we endow LLMs with the ability to self-reflect and self-evaluate? Regarding this point, many works have explored whether LLM can have spontaneous internal reflection ability. The author generates a large amount of step-wise feedback data through a strong LLM model, e.g., GPT4-Turbo. This is very practical.

Besides, this step-wise self-evaluate design is very practical. This type of  dataset is even more valuable than the MATH SFT data itself.

Finally, the author not only provides the main experimental results but also evaluates the self-evaluate ability of the trained model, considering indicators in dimensions such as P, R, F, and Acc. This is also relatively novel.

**Weaknesses:**

The main problems of this paper are as follows:
1. The experimental results seem not very satisfactory. In Table 1 and 2, it seems that the sampling + voting mechanism (self-consistency) is more effective. For example, in Table 1, Critic-CoT + Llama-3-70B-Instruct achieved an accuracy of 91.7, which seems not very advantageous. However, after adding Maj1@96, it is increased to 94.8, surpassing the method Iterative Refine proposed by the author. If the native Llama-3-70B-Instruct + Maj1@96 is used, an effect of 94.1 can also be achieved. Can the author explain this phenomenon?

2. The ability of LLMs for self-reflection and self-evaluation, as well as step-wise feedback, is interesting. However, the paper lacks a more in-depth analysis. For example, does LLM really possess this ability? Does the model trained through CRITIC-COT have this ability? From my perspective, I think LLM is unable to achieve self-reflection by completely relying on oneself. The author's carefully crafted step-wise feedback dataset can improve the mathematical performance of the model. But can it really achieve self-reflection? It seems that the experimental results provided by the author cannot effectively prove this point. I hope the author can provide a more solid analysis.

3. Although the author provides the code, it seems that I don't see the generated feedback dataset.

4. I don't quite understand what specific relationship the term "distant supervision" has with this paper. In the field of information extraction, the meaning of this term is clear. However, in "LLM for math reasoning", I don't quite understand the relationship between this term and this paper. It is recommended that the author modify the corresponding expression.


Miss References:

Many works have explored whether LLMs really have the ability of self-reflection:

Huang J, Chen X, Mishra S, et al. Large language models cannot self-correct reasoning yet.  ICLR24

Zhang W, Shen Y, Wu L, et al. Self-contrast: Better reflection through inconsistent solving perspectives.  ACL24

**Questions:**

As I mentioned above.
The question studied in the paper is very interesting, but some key experiments and analyses need to be further strengthened.

---

> ### Author Response · Authors · 2024-11-21
> **Response to Reviewer GCMB**
>
> Thank you for reviewing our paper and offering constructive feedback. We have prepared detailed responses to address each point of your concern.
>
> > The experimental results seem not very satisfactory.
>
> - On the one hand, Iterative Refinement could be more challenging for LLMs. To work properly, it requires the model to
>   1. Detect errors in its output,
>   2. Refine the mistakes,
>   3. Exit if no further errors are detected.
>
>   This pipeline could be more sensitive to error accumulation. Moreover, it only edits on a single example and has a limited retry, which is sample-efficient, but may not explore the solution space more actively, as self-consistency does.
>
> - On the other hand, self-consistency is a strong baseline, as it requires massive sampling. It leverages diverse reasoning paths and tries to mitigate the stochastic of a single sample. But under the method of Critic As Filter, we actively filter out problematic attempts and perform the majority vote on the more reasonable candidates, rather than equally account for all the predicted answers as the vanilla majority vote does, which further increases performance. This in turn demonstrates our model's strong ability to critique.
>
> > About the ability of self-reflection
>
> - Besides the main results,  through out-of-domain evaluation in Section 4.4 and Table 4, we find our model demonstrates generalized ability to critique and refine.
> - In terms of LLM's self-reflection ability, currently it's still an open question, and we hope our work as a valuable exploration could shed light on future studies in this area.
> - On the other hand, as long as we adequately improve the models' ability to critique, we could achieve test-time performance increase in the form of "self-reflection". As mentioned in our paper, after Critic-CoT training, the ability to critique and generate both improves, but they are not exactly identical. Notably, the critique ability can surpass the task-solving ability, allowing the model to detect errors even when it has a low probability of generating a valid solution, as the papers [1][2] also suggest. This indicates that by strengthening the model’s CoT critique ability beyond its generation capability, we can leverage this discriminative power to reject imperfect responses and achieve positive performance gains.
>
> [1] Self-critiquing models for assisting human evaluators
>
> [2] CriticBench: Benchmarking LLMs for Critique-Correct Reasoning
>
> > Data open-source
>
> Due to the large scale of our training data, we didn't upload the dataset yet. We plan to first conduct further verification of the training data quality, and then open source it after the paper is accepted.
>
> > Missing reference
>
> Thank you for pointing out the missing references. We have cited the paper "Large language models cannot self-correct reasoning yet" in Section 2.2, and we would cite the paper "Self-contrast" and several other papers in the revision.
>
> > Terms of "Distant Supervision"
>
> In information extraction, "distant supervision" means automatically generating large-scale weakly-labeled data, with the help of an existing database. And here in our paper, we try to leverage this term, since our Critic-CoT training data is also automatically constructed without human annotation. Several other works also use the term "distant supervision" in other domains beyond information extraction:
>  - Learning from Relevant Subgoals in Successful Dialogs using Iterative Training for Task-oriented Dialog Systems, EMNLP 2024 Findings
> - Learning To Recognize Procedural Activities with Distant Supervision, CVPR 2022
> - SciNLI: A Corpus for Natural Language Inference on Scientific Text, ACL 2022
> - TableLLM: Enabling Tabular Data Manipulation by LLMs in Real Office Usage Scenarios
> - LiteSearch: Efficacious Tree Search for LLMS
>
> Sorry for the confusion it brings. For better precision, we would consider using the term "weak supervision" to better describe our data construction process.
>
> **In conclusion, We hope that our responses provided address your primary concerns. If you have any additional questions or concerns, please do not hesitate to reach out.**

---

> > ### Comment · Reviewer_GCMB · 2024-11-21
> > **reply to author**
> >
> > Thanks to the author's reply. After reading the author's reply, I am even more confused.
> > 1. **Regarding distant supervision**: Distant supervision is denoted when there is no precise annotation. The dataset used by the author to generate the critique corpus already has accurate answer annotations (Line187-200). Why can it be called distant supervision?
> >
> > 2. **Regarding the experiment**, I don't quite understand what the author claimed in their reply that it is more efficient than self-consistency (i.e., sample-efficient). I think self-consistency only needs to sample 96 times and then vote. However, for the method in this paper, it also needs to evaluate 96 candidate answers. At least 192 invocation are needed? Is my understanding correct?

---

> ### Author Response · Authors · 2024-11-22
> **Reply to Reviewer GCMB**
>
> Thank you for the feedback, and sorry for the confusion. We would like to make further explanation as follows.
>
> ## 1. Regarding distant supervision
>
> > The dataset used by the author to generate the critique corpus already has accurate answer annotations (Line187-200)
>
> - The dataset we use, GSM8K and MATH, has only the ground truth of the final answer, typically a number or an expression. By comparing the prediction answer and ground truth answer, we can only determine whether the entire *multi-step attempt* is correct or not at the **outcome level**, while at the **process level**, the final answer do not provide granular information about the correctness of intermediate reasoning steps, thus the exact correctness of each step remains unknown. Therefore, we need human annotation or an automatic process to label these steps.
> - Previous work [1] manually annotated the intermediate steps, which is resource-intensive and costly. In our work, we automate the data construction process, **which weakly supervises the critique (identifying which steps are wrong and why) and refinement**, and replace the human with models like GPT-4-Turbo, which is more cost-friendly and scalable. Since our approach does not rely on precise human annotations at the intermediate level, **we categorize it as weak supervision.**
>
> [1] Let's Verify Step by Step, ICLR 2024
>
> ## 2. Regarding the experiment
>
> > I don't quite understand what the author claimed in their reply that it is more efficient than self-consistency (i.e., sample-efficient).
>
> There might be some misunderstandings regarding our experimental setups. Sorry for the confusion, and we would like to clarify our setting of "Iterative Refinement", "Self Consistency", "Critic As Filter" as follows.
>
> - In Self Consistency, we generate N=96 samples, and pick the most frequent answers.
> - In Critic As Filter, based on the N=96 samples, we additionally generate N=96 critiques, and in total 192 invocations. Then, we filter out erroneous attempts predicted by critique and keep more reasonable candidates, and select the most frequent answer among these candidates.
> - In Iterative Refinement, we follow previous works of "self-correct", and only sample **one** attempt first, then generate **one** corresponding critique: if it's predicted to be correct, then exit and output the answer, if not, generate **one** refinement, and critic it again. Compared with Self Consistency which generates a fixed number of **N=96** samples, **we believe Iterative Refinement is more sample-efficient than Self Consistency.**
>
>  To further address your concern, we further conduct statistical analysis regarding Iterative Refinement on the dataset of GSM8K, and verify its sample-efficiency:
>   - Invocations of Self-Consistency on GSM8K: 1319 * 96 = **126,624**
>   - Invocations of Iterative Refinement on GSM8K: among 1319 test cases, our Critic-CoT model predicts 274 problematic instances and iterates 1627 times (on average 5.94 rounds for each wrong case), which makes in total  1319 * 2 + 1627 * 2 = **5892** invocations, which is **21.5** times less than the number of calls of self-consistency.
>
> We hope our response would clarify the details and make it more comprehensive. If you have further questions, please do not hesitate to inform us.

---

> > ### Comment · Reviewer_GCMB · 2024-11-22
> > **Reply to Author**
> >
> > Thanks to the author for the reply.
> > - The expression "weak supervision" is more appropriate.
> >
> > Hope to see the corrections of the content we discussed before in the revised version.

---

> > > ### Author Response · Authors · 2024-11-27
> > > **Reply to Reviewer GCMB**
> > >
> > > Thanks! We have uploaded the revision, which made the following updates based on our discussion:
> > >
> > > - Change the phrase "distant supervision" to "weak supervision".
> > > - Citation of related work "Self-contrast" and a few other papers in Section 2.2.
> > > - Comparison between Iterative Refinement and Self Consistency in Appendix A.2.3.
> > > - Discussion on LLM's ability of self-reflection in Appendix A.2.4.
> > >
> > > Thank you again for the insightful and helpful advice.

---

### Official Review · Reviewer_B7Pw · 2024-11-04

**Soundness:** 3
**Presentation:** 3
**Contribution:** 2
**Rating:** 5
**Confidence:** 4

**Summary:**

This paper introduces a new method to train a CoT-style critic model. For a multi-step solution, the proposed Critic-CoT is can be used to refine the multi-step solution with step-level explanations and can also be used to filter response samples. The experiments are conducted on math and reasoning tasks such as GSM8K and StrategyQA.

**Strengths:**

+ The performance of the critic model is evaluated using F1, which was usually indirectly evaluated by the downstream tasks.
+ The critic-cot model may be used as an additional critic model for filtering purposes.

**Weaknesses:**

+ Based on the description, it seems that CoT for the critique (e.g., those in Figure 1) is not well used. Also, the formulation is a little bit confusing (e.g., l_i is +1 or -1, however, the cot part is not included in this representation). The refinement ability of the CoT-critic seems to be weak on challenging math tasks such as MATH and MATH500, as the improvements in these tasks are marginal. Though the authors claim the main difference is the CoT ability of the critic model compared with other types of critic models, the advantage of using CoT is not well justified.
+ The role of distillation: the revised step also can be regarded as re-writing answers (the quality may be better than the original human annotation due to factors such as consistency), it is somehow unclear the gains from in-domain data augmentation or from the critic-cot.
Considering the fact that 10 samples are leveraged to form the data per instance, labeled by the gpt-3.5, the process can be regarded as that best-of-10 responses are used for SFT. Using 16 retry gpt-4-Turbo to obtain the refinement data also brings the same issue.
+ The evaluation lacks strong baselines that also use NL critic models or other types of reward models (e.g., deterministic), PRM or ORM.
+ Some details are missing. Please see the question part.

**Questions:**

+ How to define one step?
+ How to identify pred from the attempt? I can understand for GSM8K and MATH, there may exist answer patterns. Then how can we adapt the proposed method to other tasks, in which the answers cannot be easily extracted? To use the critic-cot for filtering, majority vote is also leveraged, which also assumes that the answers can be easily extracted (or only numbers).
+ For the 2nd assumption (if the final answer is correct, then all the intermediate steps are correct), could the authors manually check tens of examples and provide the percentage of instances with perfect CoT reasoning? Also for the 1st assumption, it may also depend on the ability of the LLM.

---

> ### Author Response · Authors · 2024-11-21
> **Response to Reviewer B7Pw**
>
> Thank you for reviewing our paper and offering constructive feedback. We have prepared detailed responses to address each point of your concern.
>
> > W1&W3: Based on the description, it seems that CoT for the critique (e.g., those in Figure 1) is not well used.
>
> Whether CoT is beneficial for critique is indeed worth investigating. In Section 4.5 and Table 5, we have conducted the ablation study with the two settings, and demonstrate the necessity of CoT:
> 1. Process Label: remove the CoT critique and only let the model predict if the step is correct
> 2. Outcome Label: let the model directly predict the whole attempt is correct
>
> We find that the accuracy of critique drops. On GSM8K, our Critic-CoT model is 92.3%, while the Process Label without CoT is 89.5%, and the Outcome Label is 88.0%. Similarly, on MATH, the critique accuracy drops from 72.2% to 63.8% with Process Label, and further 63.0% with Outcome Label.
>
> **This indicates the effectiveness of CoT for critique. Also, we hope these two baselines could serve as a NL version of approximal PRM / ORM.** We would refine our presentation and more explicitly justify the necessity of CoT for critique.
>
> > W2: it is somehow unclear the gains from in-domain data augmentation or from the critic-cot.
>
> As discussed in Sections 4.3 and 4.5 of our paper, the improvements can be attributed to two key factors: training with critique and refinement, and application of critique and refinement at test time.
> - Strengthening the ability to critique and refinement would not compromise the task-solving performance, but improve it. Therefore, our critic model already exhibits noticeable performance improvements in task-solving, even in the absence of additional critique steps during the inference phase.
> - At the inference phase, we can actively leverage the model’s ability to reflect on its reasoning and correct mistakes via Critic As Filter and Iterative Refinement, which leads to additional improvements.
>
> > Q1: How to define one step?
>
> For task-solving, we instruct the model to present the attempt in a step-by-step manner, and the model would decide how long a step is, usually a sentence or a short paragraph. For example, in Figure 3, "Let the three consecutive integers be x-1, x, and x+1. The sum of these three integers is 3x." is a step.
>
> > Q2: How to identify pred from the attempt?
>
> In the original MATH dataset, they wrap the final answer with "\boxed{answer}". As the prompts in Appendix A.6 demonstrate, we let the model print the answer in the format "\boxed{answer}", then the model would write down the answer in this latex expression. We extract the regular expression \boxed{.*} from the model output, and obtain the valid answer expression with matched parenthesis. We would add Python code for answer extraction in the Appendix in revision.
>
> ```python
> import re
>
> def extract_boxed_expressions_custom(text):
>     stack = []
>     current_expr = ""
>     i = 0
>     while i < len(text):
>         if text[i:i+7] == r"\boxed{":
>             if stack:
>                 current_expr += text[i]
>             stack.append("{")
>             i += 7
>         elif text[i] == "{" and stack:
>             stack.append("{")
>             current_expr += text[i]
>             i += 1
>         elif text[i] == "}" and stack:
>             stack.pop()
>             if stack:
>                 current_expr += text[i]
>             else:
>                 current_expr = re.split('=', current_expr)[-1]
>                 return current_expr
>             i += 1
>         elif stack:
>             current_expr += text[i]
>             i += 1
>         else:
>             i += 1
>     return ''
>
> def extract_answer(text):
>     predict_answer_span = re.findall(r'(\\boxed\{.*\})', text, re.DOTALL)[-1]
>     predict_answer = extract_boxed_expressions_custom(predict_answer_span)
>     return predict_answer
> ```
>
> > Q3: Could the authors manually check tens of examples and provide the percentage of instances with perfect CoT reasoning?
>
> - To address your concern, we each sample 100 correct answer solutions, on GSM8K by GPT-3.5-Turbo and GPT-4-Turbo, and MATH by GPT-4-Turbo, then manually check if all intermediate steps are correct.
>   - The results are demonstrated in the following table. **We find that in general, the assumption holds**: Despite the presence of a small amount of noise, the data generated in this manner can simultaneously ensure both quality and scalability. Also, from GSM8K to MATH, as the reasoning traces become longer and more complicated, the percentage of correct answer but with wrong intermediate steps increases.
>     | Data | Model | Intermediate Accuracy of Correct Answer Attempt |
>     | --- | --- | --- |
>     | GSM8K | GPT-3.5-Turbo | 97% |
>     | GSM8K | GPT-4-Turbo | 99% |
>     | MATH | GPT-4-Turbo | 93% |
>
> **In conclusion, we hope that our responses address your concerns, and we will revise our manuscript accordingly. Please do not hesitate to inform us of any additional concerns or questions you may have.**

---

> ### Author Response · Authors · 2024-12-02
> **To Reviewer B7Pw**
>
> Dear Reviewer,
>
> As the rebuttal phase nears its conclusion, we have diligently provided detailed responses to address your previous concerns. We hope that you would review our replies and identify any remaining concerns or additional questions you might have. We recognize your busy schedule, but if possible, we kindly request that you share any further inquiries or feedback you may have. We are eager to further discuss and clarify any points as needed.
>
> Thank you very much for your valuable time and assistance.
>
> Best Regards,
>
> The Authors

---

> ### Author Response · Authors · 2024-12-02
> **To Reviewer B7Pw**
>
> Dear Reviewer B7Pw:
>
> Thank you again for the insightful feedback. We would like to take this opportunity to briefly revisit and clarify the key concerns you raised. Specifically,
> - The role of CoT in critique and the comparison between PRM and ORM: In Section 4.5 (Line 479-518) and Table 5, we have conducted an ablation study comparing the NL versions of "PRM" and "ORM." and the results confirm the effectiveness of stepwise CoT critique.
> - Source of improvement: We now add a discussion in Appendix A.2.2 (Lines 895-905) regarding the factors contributing to performance improvement. The two factors include:
>   - Critic-CoT training
>   - The inference methods, namely Critic As Filter and Iterative Refinement during test-time.
> - Process Correctness of Correct Answer Attempt: We now add the manual evaluation of the intermediate steps correctness of attempts with the correct answer in Appendix A.3 (Line 945-950) and Table 7. We find that in general, the correct final answer is a strong indicator of accurate CoT reasoning.
> - Identifying the prediction answer: We extract the predicted answer using the pattern \boxed{.*} from the model’s output. We now add the details and the corresponding Python code in Appendix A.4 and Table 9.
> - Definition of a step: As mentioned in our previous response,
> > For task-solving, we instruct the model to present the attempt in a step-by-step manner, and the model would decide how long a step is, usually a sentence or a short paragraph. For example, in Figure 3, "Let the three consecutive integers be x-1, x, and x+1. The sum of these three integers is 3x." is a step.
>
> As the rebuttal period is drawing to a close, we would appreciate your feedback on whether our responses sufficiently address your concerns. If there are any remaining issues, we would be happy to provide further clarifications before the end of the rebuttal period.
> Thank you once again for your time and consideration.

---

> > ### Author Response · Authors · 2024-12-03
> > **A kindly request for your response**
> >
> > Dear Reviewer B7Pw,
> >
> > We hope this message finds you well. With less than a few hours remaining in the discussion period, we kindly wanted to follow up and ensure that our recent updates have addressed your concerns. We deeply value your input and are eager to receive your feedback.
> >
> > We sincerely appreciate your time and effort in reviewing our work and would be grateful for any further comments or suggestions you might have.
> >
> > Thank you again for your attention, and we look forward to your feedback!
> >
> >
> > Best regards,
> >
> > Authors

---

### Official Review · Reviewer_xeA6 · 2024-11-04

**Soundness:** 3
**Presentation:** 3
**Contribution:** 2
**Rating:** 5
**Confidence:** 3

**Summary:**

This paper proposes a simple procedure to construct NL critic and refinement data, with those data, this paper tests two ways of using those data: iterative refine (run NL critic on a system output, and then perform refinement to produce a new output) and critic as filter (only use NL critic as re-ranking). Experiments on GSM8k and MATH test sets show some improvements when run with Iterative Refine and Critic As Filter.

**Strengths:**

1.	The data construction for NL critique at step level looks interesting.
2.	The approach of Critic As Filter shows consistent gains over majority voting.

**Weaknesses:**

1.	“Self-critique” ability used in this paper is distilled from a stronger model (e.g. GPT4-Turbo).
2.	The basic data construction in Section 3.1 is built on sampling of LLMs, this procedure will inevitably introduce some noise, the overall quality of those data must be evaluated by human and reported in this paper.
3.	It’s not clear to me where the main improvements in Table 1-3 come from, it probability comes from GPT4-Turbo annotations (supervision).

**Questions:**

1.	The quality of critique data is crucial to train a valid NL critique at step-level. A) the assumption that the final answer is correct, then all intermediate steps are correct. Please evaluate some of those data and report the accuracy. B) All Att and Cri are from sampling, that randomness will also introduce some errors, those generated data should be also evaluated and reported in paper.
2.	The Iterative Refine improvements on MATH task without re-ranking are not significant, thus, I’m not convinced self-critique refinement with single pass is a valid approach.
3.	It looks like self-critique iterative refine works in the re-ranking setting, this suggests that the self-critique accuracy is not high, we need some analysis in detail.

---

> ### Author Response · Authors · 2024-11-21
> **Response to Reviewer xeA6 (part 1)**
>
> Thank you for the thoughtful and constructive feedback. We appreciate the opportunity to clarify and address your concerns. Below, we provide detailed responses to each of your points.
>
> > W2&Q1: The quality of critique data is crucial to train a valid NL critique at step-level.
>
> - A: The assumption that the final answer is correct, then all intermediate steps are correct
>   - To address your concern, we each sample 100 correct answer solutions, on GSM8K by GPT-3.5-Turbo and GPT-4-Turbo, and MATH by GPT-4-Turbo, then manually check if all intermediate steps are correct.
>   - The results are demonstrated in the following table. **We find that in general, the assumption holds**: Despite the presence of a small amount of noise, the data generated in this manner can simultaneously ensure both quality and scalability. Also, from GSM8K to MATH, as the reasoning traces become longer and more complicated, the percentage of correct answer but with wrong intermediate steps increases.
>     | Data | Model | Intermediate Accuracy of Correct Answer Attempt |
>     | --- | --- | --- |
>     | GSM8K | GPT-3.5-Turbo | 97% |
>     | GSM8K | GPT-4-Turbo | 99% |
>     | MATH | GPT-4-Turbo | 93% |
> - B: All Att and Cri are from sampling, that randomness will also introduce some errors, those generated data should be also evaluated and reported in paper.
>   - To further address your concern, we each sampled 100 entries (50 with the correct answer and 50 with the wrong answer) from the critic-cot data on GSM8K and MATH, and conducted a manual verification to verify the accuracy of the step-wise critiques.
>   - For the critique of the correct answer attempt, it is valid if there is indeed no error in all the intermediate steps; for the critique of the wrong attempt, it is valid if the first error step and the reason for the error are both identified. A refinement is correct, if the continuation steps are flawless.
>   - The results of manual verification are demonstrated in the table, and we can see that **the data we automatically constructed maintains a high level of accuracy at the step level**, which can well support the critique training process.
>     | **Data** | **Critique of Wrong Attempt** | **Refinement of Wrong Attempt** | **Critique of Correct Answer Attempt** |
> | --- | --- | --- | --- |
> | GSM8K | 86% | 96% | 100% |
> | MATH | 84% | 96% | 94% |
> - Overall, our constructed training data, despite containing a small amount of noise, could effectively enhance the model’s critique capabilities, as evidenced by our experimental results.
> - Thanks for your valuable advice, we would conduct a more rigorous and comprehensive human validation experiment, and the corresponding results will be included in the revision.
>
> > Q2: The Iterative Refine improvements on MATH task without re-ranking are not significant, thus, I’m not convinced self-critique refinement with single pass is a valid approach.
>
> Could you clarify further on the meaning of "self-critique refinement with single pass"? Does it refer to our data construction process?
>
> If so, during the training data construction, single-pass critique refinement with retries is applicable for generating training data. By leveraging the ground truth answers as an effective filter, we can sample multiple single-pass critique-refinement pairs for each problem until success. Given the large size of problem in our dataset and there are multiple attempts at each problem, we can afford to bypass instances where valid critique-refinement pairs are not produced, and still obtain a sufficient amount of data overall.
>
> > W1: “Self-critique” ability used in this paper is distilled from a stronger model (e.g. GPT4-Turbo).
>
> Our training data of step-wise CoT critique is indeed constructed by GPT-4-Turbo via weak supervision, which is to teach the model to critique and refine its reasoning traces. **In fact, additional supervision signal is necessary due to the inadequate ability of vanilla LLMs to "self-critique".**  Previous studies like [1] have demonstrated the performance of vanilla LLMs even degrades after self-correlation. Moreover, compared to constructing data via human annotation, we automate the data construction process and replace the human with models like GPT-4-Turbo, which is more cost-friendly and scalable.
>
> [1] LARGE LANGUAGE MODELS CANNOT SELF-CORRECT REASONING YET. ICLR 2024.

---

> ### Author Response · Authors · 2024-11-21
> **Response to Reviewer xeA6 (part 2)**
>
> > W3: It’s not clear to me where the main improvements in Table 1-3 come from, it probability comes from GPT4-Turbo annotations (supervision).
>
> As discussed in Sections 4.3 and 4.5 of our paper, the improvements can be attributed to two key factors: training with critique and refinement, and application of critique and refinement at test time.
> - Strengthening the ability to critique and refinement would not compromise the task-solving performance, but improve it. Therefore, our critic model already exhibits noticeable performance improvements in task-solving, even in the absence of additional critique steps during the inference phase.
> - At the inference phase, we can actively leverage the model’s ability to reflect on its reasoning and correct mistakes via Critic As Filter and Iterative Refinement, which leads to additional improvements.
>
> > Q3: It looks like self-critique iterative refine works in the re-ranking setting, this suggests that the self-critique accuracy is not high, we need some analysis in detail.
>
> - We would like to argue that, **our model under the Critic As Filter Setting demonstrates performance increase with a large margin, which indicates the high accuracy of critique**, since the model can effectively detect erroneous attempts and filter out more reasonable candidates with a higher possibility of correctness. Additionally, as demonstrated in Table 5, on GSM8K our model has the critique accuracy of 92.3%, and 72.2% on the more challenging MATH dataset.
> - Meanwhile, in MATH dataset, even for GPT-4-Turbo, it's still difficult to refine successfully during training data construction. As mentioned in Section 4.2, we provide the reference solution to reduce intensive retry. In this setting, GPT-4-Turbo has only 78.3% accuracy on first-try refinement, 89.8% success rate under 16 retry, and on average 2.0 retry. If we remove the reference solution, GPT-4-Turbo performs much worse, with only 18.6% accuracy on first try refinement, 51.2% success rate under 16 retry, and on average 10.2 retry, which is unacceptable. Therefore, **the inherent limitation on model's refine ability constraints the iterative performance on MATH dataset.**
> - Also, as the paper [2] suggests, detecting errors could be easier than making refinements on the already sub-optimal attempts. Therefore, while refinement on hard problems may have a lower success rate, as the condition of critique is easier than generation holds, it could be more feasible to generate a batch of attempts and filter by critique. Thus, in MATH, we see Critic As Filter outperforms Iterative Refinement. We also have provided a closer inspection of Critic As Filter and Iterative Refinement in Appendix A.1.1, A.1.2, and A.1.3.
>
> [2] Self-critiquing models for assisting human evaluators
>
> **In conclusion, we hope that our responses address your concerns, and we will revise our manuscript accordingly. Please do not hesitate to inform us of any additional concerns or questions you may have.**

---

> ### Author Response · Authors · 2024-12-02
> **To Reviewer xeA6**
>
> Dear Reviewer,
>
> As the rebuttal phase nears its conclusion, we have diligently provided detailed responses to address your previous concerns. We hope that you would review our replies and identify any remaining concerns or additional questions you might have. We recognize your busy schedule, but if possible, we kindly request that you share any further inquiries or feedback you may have. We are eager to further discuss and clarify any points as needed.
>
> Thank you very much for your valuable time and assistance.
>
> Best Regards,
>
> The Authors

---

> > ### Comment · Reviewer_xeA6 · 2024-12-02
> > **Official Comment by Reviewer xeA6**
> >
> > Thanks for providing the detailed explanation.
> >
> > I will maintain my assessment and adjust the soundness and presentation ratings to better reflect the clarifications provided.

---

### Official Review · Reviewer_vkfU · 2024-11-05

**Soundness:** 3
**Presentation:** 3
**Contribution:** 2
**Rating:** 5
**Confidence:** 3

**Summary:**

This paper proposes a CoT-based critic framework, including training data collections and two different inference methods (reflection or filter).

**Strengths:**

The paper is clear, the method is well motivated.

**Weaknesses:**

At least for me, the novelty of the proposed method is limited, as it is pretty similar to existing self-reflection-based approaches.
In particular, the proposed method does not guarantee that the generated critic is reasonable, as long as the refined output (taking the generated critic as additional inputs) contains the correct answer.

Besides, I expect more experiments on broad reasoning tasks beside math, and the improvements are limited for both GSM8K and MATH.

**Questions:**

No

---

> ### Author Response · Authors · 2024-11-21
> **Response to reviewer vkfU (part 1)**
>
> Thank you for reviewing our paper and offering constructive feedback. We have prepared detailed responses to address each point of your concern.
>
> > At least for me, the novelty of the proposed method is limited, as it is pretty similar to existing self-reflection-based approaches.
>
> We would like to restate the distinction between our study and existing research, as well as highlight our contributions:
> - As we mentioned in the introduction (L42-L52), existing self-reflection works exhibit significant limitations.
>   - They are often overly simplistic, relying on basic prompts to directly point out the error, which do not incorporate deliberate step-by-step examination in completely Chain-of-Thought reasoning.
>   - They lack a targeted training process, and the ability of the vanilla LLMs to critique and refine often falls short of the demands required by complex reasoning tasks.
>   - Therefore, these limitations significantly diminish the effectiveness of self-critic and, further, self-correct. In fact, studies like [1] have demonstrated the performance of vanilla LLMs even degrades after self-correlation.
> - To address the above issues, we introduce the stepwise CoT critique format and corresponding automatic data construction framework, which is able to strengthen the model's System-2-like critic and refine abilities. Experiments also demonstrate that our approach can consistently outperform baselines without critic-CoT.  Moreover, we investigate the intrinsic correlation between critique and task-solving abilities within LLMs, discovering that these abilities can mutually reinforce each other rather than conflict.
> - **In summary, we believe our framework and findings not only differentiate our work from existing methods but also shed light on future research in this area**.
>
> [1] LARGE LANGUAGE MODELS CANNOT SELF-CORRECT REASONING YET. ICLR 2024.
>
> > In particular, the proposed method does not guarantee that the generated critic is reasonable, as long as the refined output (taking the generated critic as additional inputs) contains the correct answer.
>
> We acknowledge the importance of ensuring the quality of critiques in our framework. Given the large scale of the dataset and the high cost of intensive human annotation required for examining stepwise critiques, we currently employ weak supervision to filter and obtain more reasonable critique data while ensuring scalability. Although the process may introduce a small amount of noise, the quality of our constructed data effectively injects the model with the ability to critique and refine on a step-wise basis, thereby enhancing reasoning performance.
>
> **To further address your concern, we each sampled 100 entries (50 with the correct answer and 50 with the wrong answer) from the critic-cot data based on GSM8K and MATH, and conducted a manual verification to verify the accuracy of the step-wise critiques.**
>
> - For the critique of the correct answer attempt, it is valid if there is indeed no error in all the intermediate steps; for the critique of the wrong attempt, it is valid if the first error step and the reason for the error are both identified. A refinement is correct, if the continuation steps are flawless.
> - The results of manual verification are demonstrated in the table, and we can clearly see that the data we automatically constructed maintains a high level of accuracy at the step level, which can well support the critique training process. Thanks for your valuable advice, we would conduct a more rigorous and comprehensive human validation experiment, and the corresponding results will be included in the revision.
> | **Data** | **Critique of Wrong Attempt** | **Refinement of Wrong Attempt** | **Critique of Correct Answer Attempt** |
> | --- | --- | --- | --- |
> | GSM8K | 86% | 96% | 100% |
> | MATH | 84% | 96% | 94% |

---

> ### Author Response · Authors · 2024-11-21
> **Response to reviewer vkfU (part 2)**
>
> > Besides, I expect more experiments on broad reasoning tasks beside math
>
> **In fact, as shown in table 4, we have evaluated two out-of-domain tasks, StrategyQA and AGIEval, and the corresponding experiments demonstrate the reasoning capability of our model is generalizable.** StrategyQA is an implicit multi-hop reasoning task constructed from Wikipedia, and AGIEval contains standardized exam questions from law, english, logic, and math. Our model, especially the one trained on MATH, obtained positive performance gain on the two datasets: on StrategyQA, the accuracy increased from 76.2% (vanilla Llama-3-70B) to 80.1% through iterative refinement, and on AGIEval, the accuracy increased from 56.6% to 63.7%. This indicates the generalization beyond the training domain of math.
>
> > improvements are limited for both GSM8K and MATH.
>
> Through the training of critic-cot, the performance of GSM8K rises from 89.6% (vanilla Llama-3-70B) to 93.3% through iterative refinement, and similarly from 51.1% to 56.6% on MATH. We believe this performance improvement to be significant. As discussed in Sections 4.3 and 4.5 of our paper, the improvements can be attributed to two key factors:
> - Strengthening the ability to critique and refinement would not compromise the task-solving performance, but improve it. Therefore, our critic model already exhibits noticeable performance improvements in task-solving, even in the absence of additional critique steps during the inference phase.
> - At the inference phase, we can actively leverage the model’s ability to reflect on its reasoning and correct mistakes via Critic As Filter and Iterative Refinement, which leads to additional improvements.
>
> Furthermore, we will revise the result tables to clearly emphasize these performance gains.
>
> In conclusion, we hope our responses address your concerns regarding contribution, critique quality, generalization, and observed improvements. We are committed to revising our manuscript to incorporate these clarifications and enhancements. Please do not hesitate to share additional feedback or questions.

---

> > ### Comment · Reviewer_vkfU · 2024-11-24
> > **Limited novelty and improvements**
> >
> > ## Limited novelty
> > From my perspective, the central idea of find the erroneous step and make correction is similar with Reflexion. Though Reflexion is cited, no comparison and discussion are made.
> >
> > ## Limited improvements
> > I mentioned "limited improvements" because GPT4-turbo (instead of the llama3 70b policy) is used as the critic to provide feedbacks to the policy. This is essentially distillation.
> > Looking into MATH results, GPT4-turbo shows an accuracy of 72. Using such critic and up to 16 iterations of refinement yields an accuracy of 56.6 from 51.1.
> > This does not look exciting to me.

---

> > > ### Author Response · Authors · 2024-11-27
> > > **Response to reviewer vkfU**
> > >
> > > Thanks for the reply. We would like to provide a further response to address your concerns as follows.
> > >
> > > ## 1. Comparison Between Reflexion
> > >
> > > > the central idea of find the erroneous step and make correction is similar with Reflexion. Though Reflexion is cited, no comparison and discussion are made.
> > >
> > > Thank you for the valuable suggestion. We adopt a similar approach to Relexion [1], which leverages natural language critique to facilitate refinement, but our method diverges in the following ways:
> > >
> > > - Step-wise CoT Critique: Reflexion translates and augments the binary reward signal from the environment to natural language, but **on an instance level**. Instead, fine-grained Chain-of-Thought analysis at the step level, which is more systematic, and enables us to locate the error and start refinement **from a specific step**, rather than making broad corrections to the entire attempt.
> > > - Enhanced Critic ability: While Relexion proposed an in-context learning pipeline for optimization **under the oracle success/fail binary feedback signal**, [2] showed that without external feedback, vanilla LLMs cannot self-correct effectively due to limited critique ability. In our work, we address this limitation by training the model to develop intrinsic self-critique abilities through Critic-CoT training, and can apply it to test-time situations where the oracle feedback is not available.
> > >
> > > We believe our work pushes a step forward in the research of LLM's self-reflection. We would add the comparison and discussion in our revision.
> > >
> > > [1] Reflexion: language agents with verbal reinforcement learning, NeurIPS 2023
> > >
> > > [2] Large Language Models Cannot Self-Correct Reasoning Yet. ICLR 2024.
> > >
> > > ## 2. On Improvement
> > >
> > > Thank you for the clarification, and we would like to emphasize several key aspects of our contribution.
> > >
> > > Current SoTA LLMs like GPT-4-Turbo show remarkable performance and emergent abilities that vanilla smaller language models do not possess, but require extensive resources and considerable inference cost [3]. Therefore, whether it's possible to inject these emergent abilities into smaller models is worth investigating, as they are relatively more resource-efficient and accessible [4] [5]. However, in terms of improving math ability, current approaches to distillation mainly involve creating extremely large-scale synthetic data of *new questions* [5] [6], and may suffer from overfitting [7].
> > >
> > > On the other hand, in our work, we focus on critique, which is another emergent ability [8]. While previous works [9] applied extensive human annotation, we instead automatically apply rejection sampling only on *in-domain* training instances. As Table 3 shows, our model surpasses DART [6], which also trained Llama3-70B but on 590k examples. Compared with a concurrent work [10] that constructs a refinement dataset but without step-wise CoT critique, and improves Meta-Llama-3-70B-base from 45.7% to 46.2%, our method is also more advantageous.
> > >
> > > Finally, the main goal of our work is to demonstrate it's possible to actively train LLMs and inject critique ability, then perform intrinsic self-correct with a non-negative performance increase. We hope our work could benefit future studies in the areas of self-correction and System-2-like reasoning.
> > >
> > >
> > > We sincerely thank you for your valuable feedback and would incorporate these clarifications into our revision.
> > >
> > > [3] Efficient Large Language Models: A Survey, TMLR 2024
> > >
> > > [4] Phi-2: The surprising power of small language models
> > >
> > > [5] Small Language Models: Survey, Measurements, and Insights
> > >
> > > [6] DART-Math: Difficulty-Aware Rejection Tuning for Mathematical Problem-Solving
> > >
> > > [7] A Careful Examination of Large Language Model Performance on Grade School Arithmetic, NeurIPS 2024
> > >
> > > [8] CriticBench: Benchmarking LLMs for Critique-Correct Reasoning, ACL 2024 Findings
> > >
> > > [9] Self-critiquing models for assisting human evaluators
> > >
> > > [10] S3C-MATH: Spontaneous Step-level Self-correction Makes Large Language Models Better Mathematical Reasoners

---

### Comment · Area_Chair_2Yfp · 2024-11-25
**Reviewer response**

Dear Reviewers,

The rebuttal discussion period is coming to a close and the paper currently has a mix of positive and negative reviewers. The authors have spent a lot of time responding to each concern -- can you take a look at the author responses and let them know any remaining concerns you have?

Best,
AC

---

### Comment · Area_Chair_2Yfp · 2024-11-25

Dear Reviewers,

The rebuttal discussion period is coming to a close and the paper currently has a mix of positive and negative reviewers. The authors have spent a lot of time responding to each concern -- can you take a look at the author responses and let them know any remaining concerns you have?

Best,
AC

---

### Comment · Area_Chair_2Yfp · 2024-11-25

Dear Reviewers,

The rebuttal discussion period is coming to a close and the paper currently has a mix of positive and negative reviewers. The authors have spent a lot of time responding to each concern -- can you take a look at the author responses and let them know any remaining concerns you have?

Best,
AC

---

### Author Response · Authors · 2024-11-27
**Official Comment by Authors**

We thank all reviewers for your detailed reviews. We have uploaded the revision, which is highlighted in blue color:

- Replacement: Change the phrase "distant supervision" to "weak supervision", as suggested by Reviewer GCMB
- Addition: Citation of related work "Self-contrast" and a few other papers in Section 2.2,  as suggested by Reviewer GCMB
- Addition: Comparison between our method and Reflexion in Appendix A.2.1, as suggested by Reviewer vkfU
- Addition: Discussion on the two factors of performance improvement in Appendix A.2.2, as suggested by Reviewer xeA6, B7Pw
- Addition: Comparison between Iterative Refinement and Self Consistency in Appendix A.2.3, as suggested by Reviewer GCMB
- Addition: Discussion on LLM's ability of self-reflection in Appendix A.2.4, as suggested by Reviewer GCMB
- Addition: Manual Evaluation of the intermediate steps correctness of attempts with the correct answer, and the validity of critique and refinement in our constructed dataset, in Appendix A.3, Table 7, and Table 8, as suggested by Reviewer vkfU, xeA6, and B7Pw
- Addition: The process of answer extraction and the corresponding Python code in Appendix A.4 and Table 9, as suggested by Reviewer B7Pw

---

> ### Comment · Reviewer_GCMB · 2024-11-27
> **reply to author**
>
> Thanks to the author for his efforts to clarify.
> Generally speaking, I like this idea of critic-cot. Hoping the author can include more reasoning-related benchmarks later, such as code.

---

### Comment · Reviewer_GCMB · 2024-11-27
**reply to author**

Thanks to the author for the clarification and modification. I have increased my score.

---

### Meta-Review · Area_Chair_2Yfp · 2024-12-23

**Metareview:**

This paper proposes Critic-CoT, a method designed to improve LLMs' reasoning by developing a stepwise, chain-of-thought (CoT) critique process. The approach improves LLMs to critique and refine their solutions automatically. Specifically, the authors generate step-wise feedback data by extensive sampling and using a strong LLM (gpt-4-turbo), and train LLMs with the data to gain self-critic ability. Experiments on GSM8k and MATH test sets show certain improvements by using the self-critic ability for iterative refinement or for filtering. A common concern of the results is that it’s hard to know if the improved performance is indeed due to distilling from GPT-4-turbo, or really from the proposed Critic-CoT procedure. It’d also be good to study beyond just math (GSM8K and MATH) datasets.

**Additional Comments On Reviewer Discussion:**

The authors responded to each of reviewers' comments. The common concern among three of the reviewers seem to still remain, namely, it's hard to know if the improved performance is indeed due to distilling from GPT-4-turbo, or really from the proposed Critic-CoT procedure. If it's the former, does the model learn the self-critic ability in general domains, or is the ability specific to math problems (on GSM8K and MATH)? How would this approach compared to other distillation approaches?

---

### Decision · Program_Chairs · 2025-01-22

Reject